# It Helps to Take a Second Opinion:
# Teaching Smaller LLMs to Deliberate Mutually via Selective Rationale Optimisation

*Sohan Patnaik, *Milan Aggarwal, Sumit Bhatia, & Balaji Krishnamurthy
Media and Data Science Research Lab, Adobe
{soha, milaggar, sumit.bhatia, kbalaji}@adobe.com

## Abstract

Very large language models (LLMs) such as GPT-4 have shown the ability to handle complex tasks by generating and self-refining *step-by-step* rationales. Smaller language models (SLMs), typically with $< 13B$ parameters, have been improved by using the data generated from very-large LMs through knowledge distillation. However, various practical constraints such as API costs, copyright, legal and ethical policies restrict using large (often opaque) models to train smaller models for commercial use. Limited success has been achieved at improving the ability of an SLM to **explore** the space of possible rationales and **evaluate** them by itself through self-deliberation. To address this, we propose **COALITION**, a trainable framework that facilitates interaction between two **variants** of the same SLM and trains them to **generate** and **refine** rationales optimized for the end-task. The variants exhibit different behaviors to produce a set of diverse candidate rationales during the generation and refinement steps. The model is then trained via Selective Rationale Optimization (SRO) to prefer generating rationale candidates that maximize the likelihood of producing the ground-truth answer. During inference, COALITION employs a controller to select the suitable variant for generating and refining the rationales. On five different datasets covering mathematical problems, commonsense reasoning, and natural language inference, COALITION outperforms several baselines by up to $5\%$. Our ablation studies reveal that cross-communication between the two variants performs better than using the single model to self-refine the rationales. We also demonstrate the applicability of COALITION for LMs of varying scales (4B to 14B parameters) and model families (Mistral, Llama, Qwen, Phi). We release the code for this work here.

## 1 Introduction

Modern large language models (LLMs) with hundreds of billions of parameters, such as GPT-4 (Achiam et al., 2023) and PaLM-540B (Chowdhery et al., 2022) have shown a remarkable ability to solve complex tasks by generating step-by-step rationales (Wei et al., 2022a;b; Kojima et al., 2022) and refining them through self-correction (Wang et al., 2023b; Welleck et al., 2023). The ability to *think step-by-step* becomes more prominent with scale, while smaller language models (SLMs), typically $\lesssim 13B$, struggle to generate good quality rationales (Valmeekam et al., 2022; Weng et al., 2023). However, owing to the advantages of SLMs such as lesser costs, latency, and compute requirements, significant efforts have been made to improve their ability to handle complex tasks by using feedback obtained through interactions with LLMs (Tunstall et al., 2023; Hsieh et al., 2023; Gou et al., 2024; Wang et al., 2024b). While such approaches are suitable for research and academic settings, lack of transparency in the training data of larger (often opaque) LMs limits their use in commercial settings owing to legal, ethical and copyright concerns. For instance, OpenAI's usage terms prohibit using GPT-generated outputs to train other models for commercial use.

Consequently, efforts have been made to improve SLM performance without reliance on an external teacher LLM. In the absence of supervision from external models, the SLM has to rely on its intrinsic

---

*equal contribution

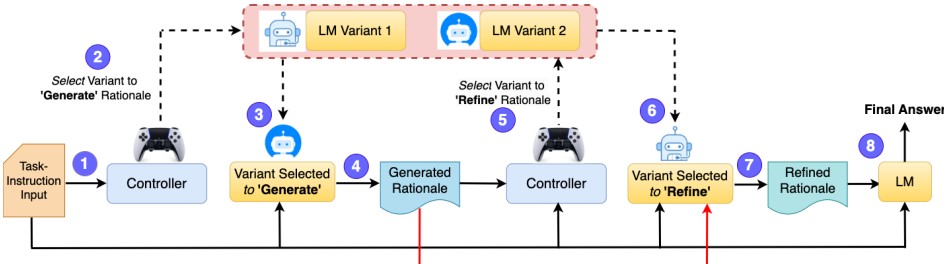

Figure 1: Schematic flow of inference using COALITION which leverages two variants of the same LM. The sample is fed to a controller (step 1) to select the variant (steps 2-3) that generates a rationale (step 4). The generated rationale is then fed to the controller to select the variant (steps 5-6) to refine the rationale (step 7) that can be used to obtain the final answer (step 8).

knowledge to explore and refine (Madaan et al., 2023a) the space of possible *reasoning paths* (Hao et al., 2023). However, due to limited scale and exploration capabilities, small models get trapped in redundant reasoning paths (Valmeekam et al., 2023; Qi et al., 2024). Furthermore, commonly used techniques such as prompt-based cross-communication between multiple LLMs for iteratively refining diverse reasoning paths (Mousavi et al., 2023; Yin et al., 2023) or using LM-as-a-judge paradigm to identify high-quality rationales to facilitate iterative refinement of rationales (Yuan et al., 2024), fail to generalize in case of smaller models (as we show empirically in § 4). While training the SLMs using task-specific ground-truth (GT) rationales has shown promise (Chen et al., 2024), the lack of availability of GT rationales for a given task limits the applicability of such methods.

With this background, the key problem that we study is how to train the SLMs without relying on external LLMs and task-specific GT rationales to (i) generate (and, refine) diverse rationales, and (ii) select the high-quality rationales leading to improved performance on end-tasks. We posit that the following two abilities **(A)** are critical to achieve this - **(A1)** ability to obtain distinct rationale candidates describing varied reasoning paths and diverse opinions about how to refine them; and **(A2)** ability to discriminate high-quality rationales from the low-quality ones to enable the model to prefer generating candidates which are more useful. Driven by this intuition, we propose **COALITION** (TeaCching LLMs tO DeliberAte MutuaLly via SelectIve RaTIonale OptimisatiON), a trainable framework that facilitates interaction between two distinct *variants* of the same SLM to learn to selectively **Generate** and **Refine** better rationale choices guided by the performance of the end-task.

The key intuition in COALITION is to overcome the limited ability of SLMs to generate diverse and high-quality rationales by employing different *variants* of the same LM that are designed to exhibit distinct behavior by optimising them on separate data splits (§ 3.1). During training, different rationales generated by the variants are further refined by each variant through self and cross-refinement. The generated and refined rationale candidates are assigned a *utility score* by estimating the likelihood of generating the final GT answer conditioned on the rationale in input. The LM variants are then tuned via preference optimization (DPO) (Rafailov et al., 2023) to prefer generating rationale candidates with higher utility scores (§ 3.2). A typical inference step in COALITION is illustrated in Figure 1 where a trained controller module (§ 3.3) is employed for selecting the appropriate model variant for generating and refining the rationale before using it for answer generation.

Empirically, we demonstrate the effectiveness of COALITION across five different datasets covering mathematics problem-solving (GSM8K (Cobbe et al., 2021)), natural language inference (PIQA (Bisk et al., 2020) and WinoGrande (Sakaguchi et al., 2020)) and commonsense reasoning (CSQA (Talmor et al., 2019) and HellaSwag (Zellers et al., 2019)). Using Llama3-8B as the base model and without any supervision from external stronger models, COALITION leads to absolute gains of up to $5\%$ over several recent baselines (§ 4.1). We also demonstrate the efficacy of COALITION for different language model families such as Phi3, Qwen 1.5, and Mistral across varying parameter-scales ranging from $4B$ to $14B$ (§ 4.2). We present results that offer evidence that cross-communication between the two variants performs better than always using a single model to self-refine the rationales (§ 4.3). Finally, we conduct extensive ablation studies to guide various design choices such as 1) the use of distinct model variants to obtain diverse rationales over sampling-based decoding through a single model and 2) task-guided rationale selection (§ 4.4).

## 2 RELATED WORK

**Prompt-Driven Reasoning Generation:** Very large-scale LLMs have been made to elicit reasoning chains by asking to generate step-by-step rationales via Chain-of-Thought (CoT) prompting (Wei et al., 2022a; Kojima et al., 2022; Wei et al., 2022b). Subsequently, several works have attempted to generate better reasoning through in-context learning (Li & Qiu, 2023a;b) by improving the quality of exemplar rationales in the prompt (Zhang et al., 2023; Diao et al., 2024). On the other hand, self-correction methods (Madaan et al., 2023a) prompt the LLM to iteratively refine its rationales using its own feedback (Welleck et al., 2023; Wang et al., 2023a). However, it has been shown that LLMs are unable to revise their own outputs without external feedback (Jiang et al., 2024b; Valmeekam et al., 2023; Stechly et al., 2023; Huang et al., 2024) owing to the fact that using the same internal representations for refinement yields redundant or incorrect reasoning paths (Yin et al., 2023).

**Performance Enhancement using External LLMs**: Various methods have explored improving an LLM by facilitating interaction with other LLMs (Jiang et al., 2023b; Yu et al., 2024; Juneja et al., 2023; Ulmer et al., 2024; Lu et al., 2024b). Exchange-of-Thought (EoT) (Yin et al., 2023) mimic the way humans conduct discussions by enabling multiple LLMs to critic (Mousavi et al., 2023) and refine each other's outputs via prompting. Our experiments show that such methods work well only with larger LLMs. Other methods distil information from a larger LM into a smaller one (Hsieh et al., 2023; Kang et al., 2023) or personalise the feedback of teacher LLM based on weaknesses of student LLM (Wang & Li, 2023; Saha et al., 2023; Jiang et al., 2023c). However, it is often argued that training over GPT-generated outputs makes smaller LLM imitate just the style (Gudibande et al., 2024) but not learn the reasoning process (Mukherjee et al., 2023). Some methods train the LLM to prefer generating certain outputs (Zhang et al., 2024a) over others via preference optimisation (DPO) (Rafailov et al., 2023). Mixture-of-Agents (Wang et al., 2024a) employs multiple open-source LLMs based agents and comprises of multiple layers of such LLM agents such that responses generated by agents in a layer are fed to LLM agents in the subsequent layer to refine the output. COALITION creates multiple variants of same SLM without involving any external LLM.

**Improving LLM Rationales through Self-Play:** Some works improve an LLM by using it to explore the reasoning space and discriminate between outputs (Qu et al., 2024; Tian et al., 2024) by itself. Tree-of-Thought (ToT) (Yao et al., 2023) organises candidates for each intermediate reasoning step in the form of a tree to look-ahead to gauge the quality of initial steps and backtrack accordingly. Zhang et al. (2024b) tunes the LLM on candidates obtained using ToT prompting through DPO. Other methods leverage sampling-based decoding (Wang et al., 2024d; Zhang et al., 2024a; Pang et al., 2024) to generate varied outputs. Such works rely on the scale of very-large LLMs to obtain diverse responses and fail to generalise well using smaller LLMs (as shown in experiments). Likewise, other works use very-large LLMs to rate the quality of different candidates for preference optimisation (Yuan et al., 2024; Pang et al., 2024) via LLM-as-a-judge (Zheng et al., 2023). However, SCORE (Zhang et al., 2024c) shows that smaller LMs need superior LLMs to verify responses for correction (Jiang et al., 2024b). On contrary, we leverage LLM's likelihood of generating final ground-truth answer conditioned on the rationale as a measure of its quality (Wang et al., 2024c). Some methods align LLM's output distribution with human-labelled data (Lai et al., 2024). SPIN performs DPO by selecting the ground-truth answer over the LLM-generated response (Chen et al., 2024). Lack of availability of GT rationales for a given task limits their applicability. Differently, we employ different variants of the same LLM to generate and refine diverse rationales for selection.

## 3 METHODOLOGY

**Overview of Approach**: COALITION aims to improve the ability of an SLM $\mathcal{M}$ to generate and refine reasoning chains for a given task without relying on any external model. To endow $\mathcal{M}$ with the capability to generate rationales for any input instruction, we first instruction fine-tune (IFT) $\mathcal{M}$ using the Chain-of-Thought (CoT) (Kim et al., 2023) dataset. The CoT data comprises general-domain instruction-rationale-answer triples. Further, we augment the IFT data with rationale refinement samples where the model is tuned to generate the rationale in the CoT data sample given the model-generated rationale in the input. This augmentation teaches the model to generate and refine rationales. Recall from the earlier discussion (§ 1) that a typical SLM has a limited capability to explore and self-correct diverse reasoning paths. Hence, we create two distinct variants of the same SLM by carrying out the IFT on separate data splits so that they exhibit different behaviors (§ 3.1).

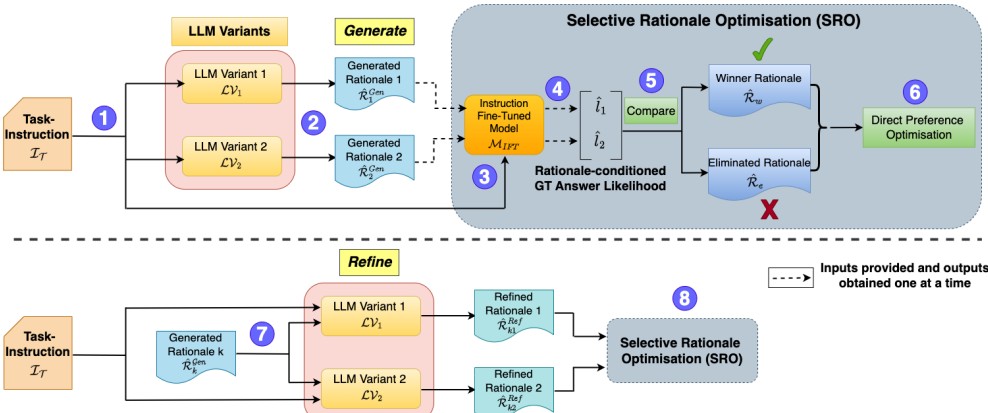

Figure 2: Training procedure of COALITION through Selective Rationale Optimisation (SRO). The task-instruction is fed to the two LLM variants (step 1) to *generate* different rationale candidates - $(\hat{\mathcal{R}}_1^{Gen}, \hat{\mathcal{R}}_2^{Gen})$. The IFT model $\mathcal{M}_{IFT}$ is used to score each candidate by estimating the likelihood $(\hat{l}_p)$ of generating the ground-truth (GT) answer conditioned on the rationale (steps 3-4). The score is used to compare the rationales to determine the winning and the eliminated rationale candidates (step 5) which are used to tune the LLM through DPO (step 6). During the *refine* stage, a generated rationale candidate $(\hat{\mathcal{R}}_k^{Gen})$ is fed to both the variants to refine the rationale (step 7). The corresponding refined rationale candidates $(\hat{\mathcal{R}}_{k1}^{Ref}, \hat{\mathcal{R}}_{k2}^{Ref})$ are used to tune the model via SRO (step 8).

Subsequently, given an end-task without explicit rationale annotations, COALITION facilitates interaction between the two variants to generate and refine the rationales. The resulting set of diverse rationale candidates are then used for task-guided Selective Rationale Optimization (SRO) (§ 3.2) to tune the variants to prefer rationales that can lead to an improved end-task performance.

Figure 2 presents the details of various components and steps involved in COALITION illustrating rationale generation, refinement, and tuning of LM variants via SRO. Specifically, given a task-specific instruction sample, it is fed to each variant of the LLM separately with the prompt to **generate** a rationale describing the steps to derive the final answer (steps 1-2 in fig. 2). Subsequently, the rationales generated by each variant are fed again to both the variants for **self-refinement** and **cross-refinement**. Owing to the distinct behavior of the variants, we obtain a set of diverse rationales at both *generate* and *refine* steps. A utility score is assigned to each rationale candidate by estimating the likelihood of generating the GT answer by the IFT model conditioned on the rationale in input (steps 3-4 in fig. 2). The candidates are ranked based on the utility score for tuning the variant LMs via DPO (Rafailov et al., 2023) (A.1) to prefer to output those generated and refined rationale choices with higher scores (steps 5-6 in fig. 2). Note that the rationale candidates for *generate* and *refine* steps are ranked and used separately for DPO training (steps 7-8 in fig. 2). We now describe the details of each component of COALITION in the subsections to follow.

## 3.1 MULTI-MODE INSTRUCTION FINE-TUNING (IFT) TO OBTAIN MODEL VARIANTS

Conventional IFT aims at enabling an LLM to follow a given instruction to generate an answer accordingly. However, as outlined in the approach overview, we require the IFT model $\mathcal{M}_{IFT}$ for three additional purposes - 1) generate rationale describing how to derive the final answer given the instruction ($\mathcal{I} \rightarrow \mathcal{R}$); 2) refine a rationale to improve its quality for a given instruction ($[\mathcal{I}; \mathcal{R}'] \rightarrow \mathcal{R}$); and 3) generate/estimate the likelihood of producing an answer given the instruction and rationale as input ($[\mathcal{I}; \mathcal{R}] \rightarrow \mathcal{A}$). To enable $\mathcal{M}_{IFT}$ to perform these three additional roles, we leverage a dataset $\mathcal{D}_{IFT}^{rationale}$ which comprises of samples containing instruction-rationale-answer triples. We format the samples using different prompt templates (A.3) to indicate the model about the mode in which it needs to generate the output. The model is endowed with the ability to refine rationale by tuning it to generate the rationale in the dataset sample ($\mathcal{R}$) given the LLM-generated rationale ($\mathcal{R}'$) in the input. Formally, given an instruction $\mathcal{I}$, rationale $\mathcal{R}$ and final answer $\mathcal{A}$ in a sample, we perform IFT on four types of samples via cross-entropy loss and teacher forcing (Vaswani et al., 2017):

$$\mathcal{L}_{\mathcal{I}\rightarrow\mathcal{R}} = -log\ p(\mathcal{R}_t|\ [\mathcal{P}_{\mathcal{I}\rightarrow\mathcal{R}};\mathcal{I};\mathcal{R}_{<t}],\theta_{IFT}) \tag{1}$$

$$\mathcal{L}_{[\mathcal{I};\mathcal{R}']\rightarrow\mathcal{R}} = -log\ p(\mathcal{R}_t|\ [\mathcal{P}_{[\mathcal{I};\mathcal{R}']\rightarrow\mathcal{R}};\mathcal{I};\mathcal{R}';\mathcal{R}_{<t}],\theta_{IFT}) \tag{2}$$

$$\mathcal{L}_{[\mathcal{I};\mathcal{R}]\rightarrow\mathcal{A}} = -log\ p(\mathcal{A}_t|\ [\mathcal{P}_{[\mathcal{I};\mathcal{R}]\rightarrow\mathcal{A}};\mathcal{I};\mathcal{R};\mathcal{A}_{<t}],\theta_{IFT}) \tag{3}$$

$$\mathcal{L}_{\mathcal{I}\rightarrow\mathcal{A}} = -log\ p(\mathcal{A}_t|\ [\mathcal{P}_{\mathcal{I}\rightarrow\mathcal{A}};\mathcal{I};\mathcal{A}_{<t}],\theta_{IFT}) \tag{4}$$

where, $p$ represents probability, $\mathcal{A}_t$ and $\mathcal{R}_t$ depict the $t^{th}$ token in GT answer and rationale respectively, $< t$ indicates tokens before $t^{th}$ index, $\mathcal{P}_m$ and $\mathcal{L}_m$ are the prompt format and loss function respectively for the $m^{th}$ mode; $[;]$ represents the operation to prepare LLM input after arranging the instruction, answer and/or rationale into mode-specific prompt $\mathcal{P}_m$, and $\theta_{IFT}$ is LLM parameters. For samples in IFT data mix which do not contain the rationales, only loss $\mathcal{L}_{\mathcal{I}\rightarrow\mathcal{A}}$ is applied. $\mathcal{M}_{IFT}$ is obtained by training base LLM on entire IFT data. To obtain two **LLM V**ariants ($\mathcal{LV}_1$, $\mathcal{LV}_2$), the LLM $\mathcal{M}$ is tuned on separate data splits by randomly dividing IFT dataset into two equal splits and assigning one split to each variant randomly. For an end-task without rationale annotations, variants are used to generate and refine diverse rationales in task-guided manner as discussed subsequently.

## 3.2 TASK-GUIDED SELECTIVE RATIONALE OPTIMISATION (SRO)

For a given end-task $\mathcal{T}$, we denote the corresponding dataset as $\mathcal{D}^{\mathcal{T}}$ which comprises of instruction-answer pairs of the form $(\mathcal{I}^{\mathcal{T}}, \mathcal{A}^{\mathcal{T}})$. However, note that the dataset for the given task does not contain the rationale annotations to tune the LLM. To address this, we leverage the distinct LLM variants to construct a set of diverse rationales via **Generate** and **Refine** steps. Given a task-instruction, it is given as input to each variant separately to generate a rationale. Each generated rationale is then fed to both the variants for *self-refinement* and *cross-refinement*. Quality of rationales obtained at each generate and refine step is determined based on its usefulness to enhance end-task performance i.e. likelihood of generating the ground-truth answer $\mathcal{A}^{\mathcal{T}}$. Each variant is then optimised to prefer generating better rationale candidates via Direct Preference Optimisation (Rafailov et al., 2023). The variant LLMs are tuned for numerous iterations by conducting multiple passes over $\mathcal{D}^{\mathcal{T}}$.

**Generate**: We denote the distinct variants obtained through IFT as $\mathcal{LV}_p^0$ ($p \in \{1, 2\}$). In particular, consider the $i^{th}$ iteration ($i \in \{1, 2\}$) such that the task-instruction $\mathcal{I}^{\mathcal{T}}$ is given as input to each variant - $\mathcal{LV}_p^{(i-1)}$ (obtained till last $(i-1)^{th}$ iteration) to generate rationale (steps 1-2 in fig. 2):

$$\hat{\mathcal{R}}_p^{Gen} = \mathcal{LV}_p^{(i-1)}([\mathcal{P}_{\mathcal{I}\rightarrow\mathcal{R}};\mathcal{I}^{\mathcal{T}}]\ |\ \theta_{\mathcal{LV}_p^{(i-1)}});p \in \{1, 2\} \tag{5}$$

**Refine**: Each generated rationale $\hat{\mathcal{R}}_p^{Gen}$ ($p \in \{1, 2\}$) is then fed as input to each variant $\mathcal{LV}_q^{(i-1)}$ ($q \in \{1, 2\}$) to refine the quality of the rationale as shown in following equation:

$$\hat{\mathcal{R}}_{pq}^{Ref} = \mathcal{LV}_q^{(i-1)}([\mathcal{P}_{[\mathcal{I};\mathcal{R}']\rightarrow\mathcal{R}};\mathcal{I}^{\mathcal{T}};\hat{\mathcal{R}}_p^{Gen}]\ |\ \theta_{\mathcal{LV}_q^{(i-1)}});p,q \in \{1, 2\} \tag{6}$$

where, $\hat{\mathcal{R}}_{pq}^{Ref}$ is the refined rationale produced by feeding the rationale generated by the $p^{th}$ variant (during the *generate* step) to the $q^{th}$ variant for *refinement*. The case when a variant refines its own rationale is referred to as *self-refinement* ($p = q$). Likewise, when a variant refines the rationale generated by the other variant, it is referred to as *cross-refinement* ($p \neq q$). Thus, we obtain a set of generated ($\{\hat{\mathcal{R}}_p^{Gen};p \in \{1, 2\}\}$) and refined ($\{\hat{\mathcal{R}}_{pq}^{Ref};p,q \in \{1, 2\}\}$) rationale candidates. Each rationale candidate ($\hat{\mathcal{R}}$) is assigned a utility score $\hat{l}$ (steps 3-4 in fig. 2) by estimating the likelihood of generating the ground-truth (GT) answer $\mathcal{A}^{\mathcal{T}}$ by the IFT model $\mathcal{M}_{IFT}$ conditioned on the rationale in input as: $\hat{l} = \pi_{\theta_{IFT}}(\mathcal{A}^{\mathcal{T}}\ |\ [\mathcal{P}_{[\mathcal{I};\mathcal{R}]\rightarrow\mathcal{A}};\mathcal{I}^{\mathcal{T}};\hat{\mathcal{R}}])$. Based on the utility score, the rationale candidates from *generate* and *refine* steps are ranked separately and used to tune the variants via DPO training.

**Direct Preference Optimisation (DPO)**: To maintain the distinctness in the behaviour of LLM variants, we tune them on separate data splits of $\mathcal{D}^{\mathcal{T}}$ by dividing the samples into two equal partitions and randomly assign one partition to each variant. Without loss of generality, consider $(\mathcal{I}^{\mathcal{T}}, \mathcal{A}^{\mathcal{T}}) \in$

$\mathcal{D}_k^{\mathcal{T}}$ (split assigned to $k^{th}$ variant $\mathcal{LV}_k$). Two types of samples are used to tune the variant via DPO (one sample corresponding to each *generate* and *refine* steps). For the **generate** step, we compare the rationales $\hat{\mathcal{R}}_p^{Gen}(p \in \{1,2\})$ based on their utility score $\hat{l}_p$ such that the candidate with higher score is selected as the winner rationale ($\hat{\mathcal{R}}_w^{Gen}$) and the one with lower score is referred to as the eliminated rationale ($\hat{\mathcal{R}}_e^{Gen}$). The $k^{th}$ variant $\mathcal{LV}_k^{(i)}$ is tuned (in current iteration $i$) to prefer the winner rationale over the eliminated one using the DPO loss $\mathcal{L}_{\mathcal{LV}_k^{(i)}}^{Gen}$ (steps 5-6 in fig. 2):

$$\mathcal{L}_{\mathcal{LV}_k^{(i)}}^{Gen} = -log\sigma[\beta(log\frac{\pi_{\mathcal{LV}_k^{(i)}}(\hat{\mathcal{R}}_w^{Gen})}{\pi_{\mathcal{LV}_k^{(i-1)}}(\hat{\mathcal{R}}_w^{Gen})} - log\frac{\pi_{\mathcal{LV}_k^{(i)}}(\hat{\mathcal{R}}_e^{Gen})}{\pi_{\mathcal{LV}_k^{(i-1)}}(\hat{\mathcal{R}}_e^{Gen})})] \tag{7}$$

where, $\beta = 0.1$ is a hyper-parameter to control divergence from a reference model. The previous iteration $(i-1)$ version of the variant - $\mathcal{LV}_k^{(i-1)}$ is used as the reference to obtain $\mathcal{LV}_k^{(i)}$. For the **refine** step, we consider the candidates obtained by refining the rationale generated by the $k^{th}$ variant in the first turn i.e. $\hat{\mathcal{R}}_{kq}^{Ref}(q \in \{1,2\})$. We compare them based on their utility score to identify the winner and eliminated rationales for the refine step as - $\hat{\mathcal{R}}_w^{Ref}$ and $\hat{\mathcal{R}}_e^{Ref}$ respectively. They are used to train the $k^{th}$ variant via DPO using the following loss $\mathcal{L}_{\mathcal{LV}_k^{(i)}}^{Ref}$ (steps 7-8 in fig. 2):

$$\mathcal{L}_{\mathcal{LV}_k^{(i)}}^{Ref} = -log\sigma[\beta(log\frac{\pi_{\mathcal{LV}_k^{(i)}}(\hat{\mathcal{R}}_w^{Ref})}{\pi_{\mathcal{LV}_k^{(i-1)}}(\hat{\mathcal{R}}_w^{Ref})} - log\frac{\pi_{\mathcal{LV}_k^{(i)}}(\hat{\mathcal{R}}_e^{Ref})}{\pi_{\mathcal{LV}_k^{(i-1)}}(\hat{\mathcal{R}}_e^{Ref})})] \tag{8}$$

**Likelihood-based Sample Filtration**: To ensure that high-quality samples are used to tune the variants for DPO training, we apply a filtration criteria to retain only those samples where the winning rationale ($\hat{\mathcal{R}}_w = \hat{\mathcal{R}}_w^{Gen}/\hat{\mathcal{R}}_w^{Ref}$) enhances the likelihood of generating the GT as follows:

$$\pi_{\theta_{IFT}}(\mathcal{A}^{\mathcal{T}} | [\mathcal{P}_{[\mathcal{I};\mathcal{R}]\to\mathcal{A}}; \mathcal{I}^{\mathcal{T}}; \hat{\mathcal{R}}_w]) > \pi_{\theta_{IFT}}(\mathcal{A}^{\mathcal{T}} | [\mathcal{P}_{[\mathcal{I}\to\mathcal{A}]}; \mathcal{I}^{\mathcal{T}}]) \tag{9}$$

Equation 9 compares the likelihood of generating the ground-truth by $\mathcal{M}_{IFT}$ for a given task instruction in the absence and presence of the winning rationale in the input. The sample is used for DPO training if the winning rationale enhances the likelihood compared to not using any rationale.

### 3.3 CONTROLLER-BASED LLM VARIANT SELECTION DURING INFERENCE

Given an instruction sample during inference, COALITION employs a **Controller** module to choose the variant LLM that should be used for the generate and refine steps. The controller is a small encoder-only LM $\mathcal{C}$ that is trained using the preference data collected during the DPO training based on which variant's rationale was selected. Figure 3 shows a schematic diagram depicting training procedure of the controller. Given the instruction sample $\mathcal{I}^{\mathcal{T}}$, consider the generate step such that the controller is trained using cross-entropy loss to perform a two-way classification between the variants conditioned on $\mathcal{I}^{\mathcal{T}}$ as input. The output label is determined as the variant which generated the winning rationale $\hat{\mathcal{R}}_w^{Gen}$. Likewise, corresponding to the refine step, controller $\mathcal{C}$ is conditioned on the instruction $\mathcal{I}^{\mathcal{T}}$ along with the winning rationale generated at the generate step as the input. It is trained to predict the variant that generates the winning refined rationale $\hat{\mathcal{R}}_w^{Ref}$ amongst $\hat{\mathcal{R}}_{kq}^{Ref}$ ($q \in \{1,2\}$). Once trained, $\mathcal{C}$ is used during inference to select the variant to generate rationale followed by choosing the variant for refinement conditioned on rationale obtained at generate step.

## 4 EXPERIMENTS AND EVALUATION

**Datasets:** We evaluate COALITION using five datasets belonging to three diverse reasoning task domains - **1) Maths Problem Solving** on GSM8K (Cobbe et al., 2021); **2) Natural Language Inference (NLI)** using PIQA (Bisk et al., 2020) and WinoGrande (Sakaguchi et al., 2020); and

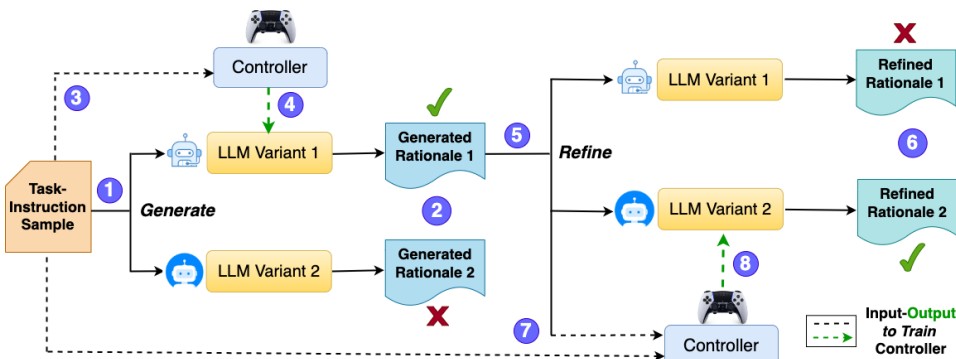

Figure 3: Training process of Controller $\mathcal{C}$. Each LLM variant *generates* a rationale candidate (step 1). The variant that generates the winning rationale (step 2) is selected as the output label for $\mathcal{C}$ (steps 3-4). For the *refine step*, $\mathcal{C}$ is conditioned on the task instruction and rationale from the *generate* step, and is trained to select the LLM variant that generates the better *refined* rationale (steps 5-8).

**3) Commonsense Reasoning** through CSQA (Talmor et al., 2019) and HellaSwag (Zellers et al., 2019). **GSM8K** comprises of maths word problems which require a model to understand the problem text and perform a sequence of calculations. **PIQA** requires understanding of physical relation between objects and comprises of samples with a goal text coupled with two candidate statements with the task of identifying the statement that can lead to the goal. **WinoGrande** is a very challenging co-reference resolution task, comprising of a statement with two parts such that the latter half refers to some entity in the first part. **CSQA** tests model's ability to answer MCQ questions by picking correct choice using commonsense knowledge. **HellaSwag** evaluates ability to predict continuation of a context by choosing most plausible ending. Number of samples in train/test splits of each dataset is GSM8k - 7.5k/1.3k, PIQA - 16k/3k, WinoGrande - 40k/1.7k, CSQA - 9.7k/1.2k and HellaSwag - 39.9k/10k. Please refer to A.2 for examples present in each dataset.

**Implementation Details:** We use a batch size (BS) of 16 on 8 80GB A100 GPUs (BS of 2/GPU), a learning rate (lr) of 1e-5, bfloat16 precision with cosine annealing (Loshchilov & Hutter, 2017) using AdamW optimizer (Loshchilov & Hutter, 2019). We leverage DeepSpeed Zero 2 with sharding of optimizer states and gradients across GPUs and enable gradient check-pointing. For Multi-Mode IFT, we experiment with different base-LLM backbones $\mathcal{M}$ (Phi3-3.8B, Qwen1.5-4B, Qwen1.5-7B, Qwen1.5-14B, Mistral-7B and LLaMA3-8B) to obtain the corresponding IFT models. We use a set of 140k samples selected randomly from CoT-Collection (Kim et al., 2023) as the data $\mathcal{D}_{IFT}^{rationale}$ to enable three additional IFT modes. Additionally, we use 40k samples from the Dolly-HHRLHF (MosaicML, 2023) and the Open Assistant datasets combined to create data for the conventional instruction-to-answer IFT mode. The IFT training is performed for 2 epochs. For task-guided SRO, we carry out 2 iterations of task-guided DPO with 10 epochs in each iteration. Controller $\mathcal{C}$ is deberta-v3-large (He et al., 2021) model trained for 30 epochs with a learning rate of 1e-5, BS of 128 (16 per GPU on 8 GPUs) using adam optimizer with cosine annealing.

**Evaluation Protocol:** To assess the usefulness of rationales, prompt containing the input-instruction is appended (during inference) with the rationale generated by a method and fed to the instruct version of the LLM. The accuracy achieved is indicative of the usefulness of rationales.

## 4.1 DOES COALITION HELP IMPROVE LLM PERFORMANCE?

Table 1 compares COALITION with two categories of **baselines**: **(I) Prompting-Techniques** to **(i)** generate rationales using **Chain-of-Thought** (Wei et al., 2022b), **(ii)** explore the space of rationales using **Tree-of-Thought** (Yao et al., 2023), **(iii)** refine the rationales using **CoT Self-Consistency** (Wang et al., 2023b) & **Self-Refine** (Madaan et al., 2023b), or **(iv)** facilitate communication between multiple LLMs to refine the rationales through **Exchange-of-Thought** (Yin et al., 2023); and **(II) Rationale Enhancement via Trainable Self-Play** - **(i) Distilling Step-by-Step** (Hsieh et al., 2023) where the IFT model is used to generate and refine the rationale, **(ii) Self-Rewarding LMs** (Yuan et al., 2024) uses sampling-based decoding to obtain diverse rationales and LLM-as-a-judge to rate their quality for DPO, and **(iii) SPIN** (Chen et al., 2024) performs DPO by

| Method | Maths GSM8K | NLI WinoGrande | PIQA | Comonsense HellaSwag | CSQA |
|---|---|---|---|---|---|
| Meta-Llama3-8B-Instruct w/o rationale (Dubey et al., 2024) | 75.89 | 71.98 | 78.51 | 57.69 | 76.17 |
| **Prompt-Driven Rationale Refinement** | | | | | |
| Chain-of-Thought (Wei et al., 2022b) | 62.39 | 60.17 | 68.48 | 45.33 | 65.28 |
| CoT Self-Consistency (Wang et al., 2023b) | 64.11 | 62.37 | 71.42 | 46.13 | 68.92 |
| Tree-of-Thought (Yao et al., 2023) | 68.11 | 70.62 | 75.14 | 53.18 | 74.37 |
| Exchange-of-Thought (Yin et al., 2023) | 69.19 | 66.47 | 73.11 | 52.22 | 75.48 |
| Self-Refine (Madaan et al., 2023b) | 77.26 | 72.81 | 79.49 | 60.48 | 78.22 |
| **Rationale Enhancement via Trainable Self-Play** | | | | | |
| Distilling Step-by-Step (Hsieh et al., 2023) | 76.18 | 70.41 | 78.77 | 56.10 | 76.31 |
| Self-Rewarding LMs (Yuan et al., 2024) | 72.16 | 68.15 | 75.22 | 55.39 | 76.15 |
| Self-Play Fine-Tuning (SPIN) (Chen et al., 2024) | 77.01 | 71.85 | 79.02 | 58.22 | 76.48 |
| **COALITION (ours)** | **81.06** | **77.13** | **83.26** | **63.23** | **82.06** |

Table 1: COALITION significantly outperforms multiple categories of baselines on 5 datasets from 3 task domains - Maths Problem Solving, Natural Language Inference (NLI) and Commonsense Reasoning. Same LLM-backbone (Llama3-8B) is used in all the methods for uniform comparison.

| Rationales Generated by Best Baseline (SPIN) and LLM Variants - $\mathcal{LV}_1$ and $\mathcal{LV}_2$ in COALITION on GSM8K Sample |
|---|
| **Instruction:** A company pays each of its employees **$600 in a month**. The company has a policy of increasing the salaries of each of its employees by **10% of the initial salary every year** for those who've **stayed in the company for five years**. If Sylvie just **clocked 5 years** in the company last December, what's her **annual** salary after **three more years** of service? |
| **Best Baseline:** Sylvie's annual salary after three more years will be **$3600** because her salary **increases by $600 every year for 5 years**. So after three years, it will be $**600*5+$600=$3600**. ✗ |
| **Generate w $\mathcal{LV}_1$:** Sylvie's salary will **increase by $600 every year for three years**, so her annual salary after three more years of service will be **$600+($600*3)=$2,400**. ✗ |
| **Generate w $\mathcal{LV}_2$:** Sylvie receives a 10% increase in her salary every year after 5 years of service. Since her current salary is $600, it will **increase by $60 each year**. After three more years, her new salary will be **$600+(3*$60)=$780**. ✗ |
| **Cross-Refine $\mathcal{LV}_2$ rationale w $\mathcal{LV}_1$:** Sylvie's initial monthly salary is $600. After completing five years, her salary increases by 10% of initial salary, making it $600*1.1=$660. After three more years of service, she will have received two additional annual 10% increases, which means her **monthly salary will be $660+$60+$60=$780**. Therefore, her **annual** salary after three more years will be **$780*12=$9360**. ✓ |
| **Self-Refine $\mathcal{LV}_2$ rationale w $\mathcal{LV}_2$:** Sylvie's salary increases by 10% each year after her initial five years. Her monthly salary after five years is $660, and in three more years, she will have received two additional 10% raises. Therefore, her monthly salary will be **$660(1.1)*(1.1)=$756**, and her annual salary will be **$756*12=$9072**. ✗ |
| **Note:** Controller selected $\mathcal{LV}_2$ for **generate** step and $\mathcal{LV}_1$ for **refine step**. Variant selection in this order yields correct rationale. |

Table 2: COALITION yields better rationale using *generate* and *refine* steps via LLM variants. Wrong and right parts in a rationale are in **red** and **green**. Baseline wrongly applies increase for first five years. $\mathcal{LV}_1$ estimates wrong annual increase while $\mathcal{LV}_2$ gives correct monthly increase but question asks annual salary. Cross-refining using $\mathcal{LV}_1$ (as selected by controller) rectifies this error.

selecting general-domain GT rationales from CoT data over LLM-generated rationales. We employ same LLM backbone (Llama3-8B) for COALITION and all baselines for a uniform comparison.

COALITION outperforms all the baselines uniformly across the three task domains (Table 1). It performs better than the best baseline (SPIN) by $\sim 4\%$ on GSM8K indicating the utility of rationales from COALITION for solving maths problems. Further, COALITION performs significantly better than SPIN for NLI with a gain of $5.3\%$ on challenging WinoGrande task and $4.2\%$ on PIQA. On commonsense reasoning, COALITION outperforms SPIN by more than $5\%$ on both CSQA and HellaSwag. Table 2 shows that COALITION generates better rationales than the best baseline (SPIN) by the virtue of employing distinct LLM variants. Please refer to A.6 for more qualitative examples.

Additionally, COALITION gives a significant performance boost compared to the case where the instruct model is evaluated without rationales, as well as using the rationales generated by the IFT model (Distilling Step-by-Step). Moreover, the baseline 'Self-Rewarding Language Models' which relies on the scale of very-large LMs to both generate and rate diverse rationales for DPO does not generalise with 8B-parameter LM. COALITION performs better than this baseline by $\sim 9\%$ on GSM8K and WinoGrande, $\sim 8\%$ on PIQA and HellaSwag, and $\sim 6\%$ on CSQA. Finally, we note that COALITION performs better by 3-4% than the best prompting-based baseline i.e. Self-Refine (Madaan et al., 2023b) which uses same LLM as generator, refiner and feedback provider.

| Model | Parameter Scale | Maths GSM8K | NLI WinoGrande | PIQA | Commonsense HellaSwag | CSQA |
|---|---|---|---|---|---|---|
| Phi3 (Abdin et al., 2024) | 3.8B | 10.36 | 73.32 | 80.30 | 59.01 | 72.48 |
| **w/ COALITION (ours)** | 3.8B | **14.76** | **76.19** | **84.48** | **63.72** | **75.04** |
| | | | | | | |
| Qwen1.5 (Bai et al., 2023) | 4B | 3.49 | 67.01 | 75.57 | 52.01 | 74.61 |
| **w/ COALITION (ours)** | 4B | **5.58** | **69.26** | **76.48** | **53.37** | **78.29** |
| Qwen1.5 (Bai et al., 2023) | 7B | 57.01 | 69.53 | 79.54 | 61.06 | 81.00 |
| **w/ COALITION (ours)** | 7B | **61.37** | **75.02** | **83.11** | **64.22** | **85.11** |
| Qwen1.5 (Bai et al., 2023) | 14B | 69.37 | 76.01 | 81.45 | 65.57 | 84.19 |
| **w/ COALITION (ours)** | 14B | **74.88** | **82.84** | **84.39** | **69.22** | **87.48** |
| | | | | | | |
| Mistral (Jiang et al., 2023a) | 7B | 48.52 | 74.43 | 81.66 | 64.78 | 69.21 |
| **w/ COALITION (ours)** | 7B | **54.42** | **78.39** | **85.01** | **68.48** | **74.38** |
| | | | | | | |
| LLaMA3 (Dubey et al., 2024) | 8B | 75.89 | 71.98 | 78.51 | 57.69 | 76.17 |
| **w/ COALITION (ours)** | 8B | **81.06** | **77.13** | **83.26** | **63.23** | **82.06** |

Table 3: Performance evaluation of COALITION with LMs of varying scale of parameters (4B to 14B) and different model families (Phi3, Qwen1.5, Mistral, Llama3). It is observed that COALITION yields significant gains on all tasks for different model families and parameter scales.

## 4.2 How Does COALITION Work With Different Model Families and Across Varying Parameter Scales?

We study if COALITION improves performance of LMs with varying scale of parameters (ranging from **4B to 14B**) and belonging to different model families. We compare the accuracy achieved using the rationales generated by COALITION vs. the rationales obtained from the IFT ($\mathcal{M}_{IFT}$) version of the LLM. Specifically, we experiment with **1) Phi3-3.8B** (Abdin et al., 2024), **2) Qwen1.5-(4B, 7B, 14B)** (Bai et al., 2023), **3) Mistral-7B** (Jiang et al., 2023a), and **4) LLaMA3-8B** (Dubey et al., 2024). Table 3 summarises the results where it can be seen that COALITION improves performance on all the tasks uniformly over different parameter-scales and LM families. In particular, consider Qwen - at 14B parameter scale, COALITION improves accuracy by $\sim 5 - 7\%$ on GSM8K and WinoGrande, and by upto $4\%$ on PIQA, HellaSwag and CSQA. For Qwen-7B, there is a similar improvement of $\sim 4 - 5\%$ on all the tasks. Likewise, performance increase is observed for Qwen-4B model. For Phi3 model comprising of 3.8B parameters, $\sim 3 - 5\%$ improvement is observed for different tasks. For Mistral-7B, there is an improvement of $5 - 6\%$ on GSM8K and CSQA, and $\sim 4\%$ on the remaining tasks. Appendix A.4 shows that COALITION is effective on general benchmarks even when trained in a task-agnostic way on open-domain samples.

## 4.3 Variant Selection via Controller and Cross-Refinement Boosts Accuracy

We analyse the usefulness of controller in Table 4 by evaluating the rationales obtained during inference - **I) w/o any refinement** (rows 1 and 2), **II) w self-refinement** using a single variant (rows 3 and 4), **III) w cross-refinement** using a fixed order of variants for all samples (rows 5 and 6). Last row corresponds to dynamic selection of the LM variants by the controller for generation and refinement that yields the best performance. In particular, generating and self-refining the rationale using a single LM variant usually (but not necessarily) performs better than not refining (rows 3-4 vs rows 1-2). However, generating the rationale using a variant and refining it with the other variant consistently performs better than self-refinement (rows 5-6 vs rows 3-4). This shows that having distinct LLM variants and taking second opinion from the other variant through cross-communication is helpful. Finally, selecting the LLM variant via the controller to generate and refine rationales based on suitability for the given sample yields significantly better results (last row).

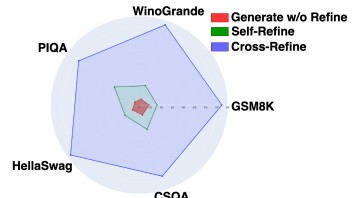

Figure 4: Proportion of samples routed by Controller to **1)** Generate w/o Refine (5-10%), **2)** Self-Refine (15-25%), **3)** Cross-Refine (65-75%). Hence, generation with one variant and refining with other is preferred mode.

| Communication Mode | Maths GSM8K | NLI WinoGrande | PIQA | Commonsense HellaSwag | CSQA |
|---|---|---|---|---|---|
| Generate (w $\mathcal{LV}_1$) w/o Refine | 77.26 | 75.10 | 80.11 | 59.22 | 78.47 |
| Generate (w $\mathcal{LV}_2$) w/o Refine | 77.21 | 74.89 | 79.24 | 59.31 | 78.33 |
| Self-Refine ($\mathcal{LV}_1 \rightarrow \mathcal{LV}_1$) | 77.35 | 74.91 | 79.86 | 59.89 | 79.35 |
| Self-Refine ($\mathcal{LV}_2 \rightarrow \mathcal{LV}_2$) | 77.23 | 74.70 | 79.60 | 59.70 | 79.25 |
| Cross-Refine ($\mathcal{LV}_1 \rightarrow \mathcal{LV}_2$) | 79.94 | 75.52 | 81.31 | 61.21 | 80.16 |
| Cross-Refine ($\mathcal{LV}_2 \rightarrow \mathcal{LV}_1$) | 79.53 | 75.83 | 81.02 | 60.87 | 80.21 |
| **COALITION (w Controller)** | **81.06** | **77.13** | **83.26** | **63.23** | **82.06** |

Table 4: Performance analysis of rationales inferred - 1) w/o refinement (rows 1-2), 2) w self-refinement (rows 3-4), 3) w cross-refinement using fixed order of variants for all samples (rows 5-6), and 4) w controller (last row). Selecting LM variants using the **controller** for the generate and refine steps yields best results. Cross-communication between the variants is better than both self-refine (using a single variant) and not refining by directly using rationale generated by a variant.

| Ablation ID | Distinct LLM Variants | Likelihood-based Rationale Selection | DPO Sample Filtration | Maths GSM8K | NLI WinoGrande | PIQA | Comonsense HellaSwag | CSQA |
|---|---|---|---|---|---|---|---|---|
| 1 | No | Yes | Yes | 71.74 | 69.28 | 76.15 | 54.22 | 75.22 |
| 2 | Yes | No | Yes | 78.24 | 75.69 | 80.14 | 60.21 | 77.49 |
| 3 | Yes | Yes | No | 75.46 | 72.28 | 76.92 | 58.55 | 77.43 |
| 4 | Yes | No | No | 73.19 | 71.37 | 76.16 | 56.92 | 77.01 |
| 5 | No | No | No | 72.16 | 68.15 | 75.22 | 55.39 | 76.15 |
| **COALITION** | Yes | Yes | Yes | **81.06** | **77.13** | **83.26** | **63.23** | **82.06** |

Table 5: Ablation study to analyse the impact of different design choices - 1) In the absence of distinct LLM variants, sampling-based decoding is performed using single LLM; 2) LLM-as-a-judge is employed when likelihood-based rationale selection is omitted; 3) Entire train set is used for DPO when sample filtration is skipped. It is observed that all components are critical for accuracy gains.

## 4.4 ABLATION STUDY - IMPACT OF DIFFERENT DESIGN CHOICES

We examine the effectiveness of following components (using Llama3-8B backbone) - 1) Distinct LLM Variants, 2) Task-guided Likelihood-based Rationale Selection, and 3) Sample Filtration for DPO. Table 5 shows the results where it can be seen that obtaining diverse refined rationales from distinct LLM variants gives significantly better performance than sampling-based decoding using a single model (**COALITION vs. row 1**). To analyse the importance of using likelihood of generating the GT answer as the utility score to rate a rationale, we leverage LLM-as-a-judge paradigm as an alternative where the IFT version of the LLM is prompted to rate rationales (**row 2**). Notable drop in performance than COALITION indicates that prompt-based rating does not work at scale of small LMs. Likewise, filtration of samples for task-guided DPO conditioned on whether the best rationale enhances the likelihood of generating GT (than not using any rationale) is critical for gains achieved by COALITION (**vs. row 3**). Additionally, omitting both likelihood-based rationale selection and sample filtration leads to further accuracy degradation (**row 4**). Finally, excluding all three components (**row 5**) gives significantly lower performance than COALITION.

## 5 CONCLUSION

We presented COALITION, a trainable framework to improve the performance of (smaller) language models on complex tasks by employing distinct variants of the same LM and use them to generate and refine high-quality diverse rationales without any supervision from external (stronger) models and ground-truth rationale annotations. The model variants are made to exhibit distinct behaviour by training them on separate data splits. In addition to the cost advantages of using smaller models, it has significant real-world advantages where legal and ethical constraints restrict the use of external models for supervision. Rigorous empirical evaluation over five datasets demonstrated the effectiveness of COALITION over several prompt-only and trainable self-play baselines. In future work, it is worthwhile to explore the impact of adding more variants on the performance and incorporating domain-specific models (specialized variants) to generate high-quality rationales.

## 6 ETHICS AND REPRODUCIBILITY STATEMENT

We employ publicly available datasets and LLMs to conduct the study in our work which are commonly used for ML research without any potential concerns. We do not annotate any data manually in this work. The rationales generated at different steps of the proposed method are of similar nature and domain as that of the text present in the datasets used. To encourage reproducibility, we release our code at this anonymous link and also upload it as part of the supplementary zip. We described the details of the datasets in § 4 (under 'Datasets' in the Experiments section) and the LLMs used in § 4 (under 'Implementation Details'). Further, we provide the implementation details of our method in § 4 (under 'Implementation Details') and discuss baselines used for comparison in § 4.1. Finally, we elaborate further details of our method in the Appendix - 1) Example of samples for each dataset (A.2) and 2) prompt templates used to format the samples during the IFT (A.3).

**Statement on Explainability:** LLMs are commonly used to generate the final answer to an input question/instruction for various NLP tasks. However, it was shown that eliciting the LLM to generate a rationale first followed by the final answer results in better accuracy. A rationale is a statement in natural language that describes the steps which are required to derive the answer, or an explanation about how the question/instruction needs to be approached to arrive at the right answer. The proposed COALITION framework improves the reasoning ability of (smaller) LLMs by improving their ability to generate better rationales. Since the rationales provide an explanation about why the LLM generated the final answer instead of just generating the final answer, the rationales can be used as a means of explainability while generating the answer to an input question/instruction. Further, since COALITION generate and refine multiple rationales using variants of the same LLM, the generated and the refined rationales can be compared to identify differences in their explanation and quality. The identified differences can provide further insights about what needs to be modified in the explanations.

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

| Dataset | Task-Instruction and Ground-Truth Answers for Each Dataset used to evaluate COALITION |
|---------|---------------------------------------------------------------------------------------|
| GSM8K | **Instruction:** Betty is saving money for a new wallet which costs $100. Betty has only half of the money she needs. Her parents decided to give her $15 for that purpose, and her grandparents twice as much as her parents. How much more money does Betty need to buy the wallet? 
 **Ground-Truth Response:** 5 |
| WinoGrande | **Instruction:** Terry tried to bake the eggplant in the toaster oven but the _______ was too big. A. eggplant, B. toaster 
 **Ground-Truth Response:** A. eggplant |
| PIQA | **Instruction:** How to dry flowers? sol1: Find a dark, moist area with good circulation, such as an attic or unused closet. With unflavored dental floss, secure the bottom of the flowers' stems to a hanger so that they hang upside down to dry. Leave flowers for two to three weeks until completely dry, sol2: Find a dark, dry area with good circulation, such as an attic or unused closet. With unflavored dental floss, secure the bottom of the flowers' stems to a hanger so that they hang upside down to dry. Leave flowers for two to three weeks until completely dry. 
 **Ground-Truth Response:** sol2 |
| HellaSwag | **Instruction:** Then he takes a small stone from the flowing river and smashes it on another stone. He starts to crush the small stone to smaller pieces. He _______________ A. cuts the center stone in half and blow it on to make it bigger. B. grind it hard to make the pieces smaller, C. eventually brings it back into view and adds it to the smaller ones to make a small triangular shaped piece, D. starts to party with them and throw the pieces by hand while they celebrate. 
 **Ground-Truth Response:** B |
| CSQA | **Instruction:** When learning about the world and different cultures, what is important if you are committed to eliminating preconceived notions. A. newness, B. loss of innocence, C. enlightenment, D. open mind, E. smartness 
 **Ground-Truth Response:** D. open mind |

Table 6: Examples of instructions from different datasets belonging to diverse task domains used in the experiments - (i) Maths Problem Solving (GSM8K), (ii) Natural Language Inference (Wino-Grande and PIQA), and (iii) Commonsense Reasoning (HellaSwag and CSQA).

# A    APPENDIX

## A.1    DIRECT PREFERENCE OPTIMISATION (DPO)

Direct Preference Optimisation (DPO) (Rafailov et al., 2023) was introduced as an alternative to Reinforcement Learning using Human Feedback (RLHF) (Ouyang et al., 2022) technique to alleviate the need of training a reward model. RLHF depends on training a reward model to assign a score to the outputs generated by an LLM to fine-tune the LLM through reinforcement learning to align it with human preferences. On the other hand, DPO transforms the loss over the reward-function to a loss over the LLM policy such that the reward is optimised implicitly by optimising the loss over the policy. It does so by leveraging human preference data which compares two possible outputs generated by an LLM such that the better output is considered as the winner candidate - $y_w$ while the inferior output is considered as the loser candidate - $y_l$. Given a static dataset of the form $\mathcal{D} = \{x, y_w, y_l\}$, where x is the input, the loss is modeled as -

$$\mathcal{L}_R = -log[\sigma(r(x, y_w) - r(x, y_l))] \tag{10}$$

$$r(x, y) = \beta log(\frac{\pi_\theta(y|x)}{\pi_{ref}(y|x)}) \tag{11}$$

where, $\pi_{\mathcal{Z}}(y|x)$ is the likelihood of generating $y$ given $x$ as input to the model $\mathcal{Z} \in \{\mathcal{M}_{ref}, \mathcal{M}_\theta\}$, $\mathcal{M}_{ref}$ is usually taken to be the instruction fine-tuned model in the case of an LLM to prevent the LLM policy from deviating too much from the initial policy, $\mathcal{M}_\theta$ represents the LLM policy being optimised through DPO, $\sigma$ is the sigmoid activation, and $\beta$ is a coefficient that controls the amount of deviation from the reference model. In summary, the algorithm optimises the LLM to learn to prefer generating certain outputs over other candidates without requiring an explicit reward model. Please refer to the original publication (Rafailov et al., 2023) for an elaborate discussion of the details.

## A.2    DATASET SAMPLES

Details of datasets were discussed in the 'Experiments' section (§ 4) in the main paper. Table 6 in the appendix shows samples of instructions for each dataset from all task domains - (i) Maths Problem Solving (GSM8K), (ii) Natural Language Inference (WinoGrande and PIQA), and (iii) Commonsense Reasoning (HellaSwag and CSQA).

## A.3    PROMPT TEMPLATES FOR MULTI-MODE INSTRUCTION FINE-TUNING

As discussed in the Methodology section (§ 3, § 3.1) in the main paper, the base model $\mathcal{M}$ is instruction fine-tuned to enable the LLM to operate in four modes in total - (i) generate the rationale given the instruction as input ($\mathcal{I} \rightarrow \mathcal{R}$); (ii) refine a rationale to improve its quality for a given

instruction ($[\mathcal{I}; \mathcal{R}'] \rightarrow \mathcal{R}$); (iii) generate the answer conditioned on the instruction and rationale as input ($[\mathcal{I}; \mathcal{R}] \rightarrow \mathcal{A}$); and (iv) generate the final answer given the instruction as input ($\mathcal{I} \rightarrow \mathcal{A}$). The inputs to the LLM are formatted using corresponding prompts ($\mathcal{P}_{\mathcal{I} \rightarrow \mathcal{R}}$; $\mathcal{P}_{[\mathcal{I}; \mathcal{R}'] \rightarrow \mathcal{R}}$; $\mathcal{P}_{[\mathcal{I}; \mathcal{R}] \rightarrow \mathcal{A}}$; $\mathcal{P}_{\mathcal{I} \rightarrow \mathcal{A}}$) for each of these modes so that the LLM can generate an appropriate output accordingly. The textual instruction for each prompt template is specified as follows:

1. $\mathcal{P}_{\mathcal{I} \rightarrow \mathcal{R}}$ = "You are an AI assistant 'M'. Provide a response to the given instruction denoted by Task Description.

   [TASK DESCRIPTION STARTS]
   $\langle$Task Description$\rangle$: In this task, you will be given an 'Instruction'. Generate descriptive reasoning on how to derive the correct answer for the instruction such that the descriptive reasoning will be useful to another AI assistant to generate the correct answer.
   'Instruction' - $\langle$instruction$\rangle$
   [TASK DESCRIPTION ENDS]

   For the given $\langle$Task Description$\rangle$, give your response. [M RESPONSE BEGINS]: "

2. $\mathcal{P}_{[\mathcal{I}; \mathcal{R}'] \rightarrow \mathcal{R}}$ = "You are an AI assistant 'M'. Provide a response to the given instruction denoted by Task Description.

   [TASK DESCRIPTION STARTS]
   $\langle$Task Description$\rangle$: In this task, you will be given an 'Instruction' and a rationale denoted by 'Rationale'. The 'Rationale' may or may not be correct for the given 'Instruction'. Analyse the rationale for its correctness, modify the rationale, and provide the correct elaborate descriptive reasoning or 'Rationale' which will be helpful to come up with the correct answer for the given instruction.
   'Instruction' - $\langle$instruction$\rangle$
   'Rationale' - $\langle$rationale$\rangle$
   [TASK DESCRIPTION ENDS]

   For the given $\langle$Task Description$\rangle$, give your response. [M RESPONSE BEGINS]: "

3. $\mathcal{P}_{[\mathcal{I}; \mathcal{R}] \rightarrow \mathcal{A}}$ = "You are an AI assistant 'M'. Provide a response to the given instruction denoted by Task Description.

   [TASK DESCRIPTION STARTS]
   $\langle$Task Description$\rangle$: In this task, you will be given an 'Instruction' and a rationale denoted by 'Rationale'. Analyse the rationale and come up with the correct answer for the given instruction.
   'Instruction' - $\langle$instruction$\rangle$
   'Rationale' - $\langle$rationale$\rangle$
   [TASK DESCRIPTION ENDS]

   For the given $\langle$Task Description$\rangle$, give your response. [M RESPONSE BEGINS]: "

4. $\mathcal{P}_{\mathcal{I} \rightarrow \mathcal{A}}$ = "You are an AI assistant 'M'. Provide a response to the given instruction denoted by Task Description.

   [TASK DESCRIPTION STARTS]
   $\langle$Task Description$\rangle$: In this task, you will be given an 'Instruction'. Generate the correct answer for the given instruction.
   'Instruction' - $\langle$instruction$\rangle$
   [TASK DESCRIPTION ENDS]

   For the given $\langle$Task Description$\rangle$, give your response. [M RESPONSE BEGINS]: "

| Model | Parameter Scale | MMLU | HellaSwag | ARC | | TruthfulQA | | WinoGrande | PIQA | GSM8k | CSQA |
|---|---|---|---|---|---|---|---|---|---|---|---|
| | | | | Easy | Challenge | MC1 | MC2 | | | | |
| Phi3 (Abdin et al., 2024) | 3.8B | 69.94 | 59.01 | 81.90 | 53.92 | 36.60 | 54.43 | 73.32 | 80.30 | 10.36 | 72.48 |
| w/ COALITION (ours) | 3.8B | **72.01** | **60.19** | **82.45** | **55.79** | **37.38** | **56.19** | **74.48** | **82.01** | **12.15** | **73.69** |
| | | | | | | | | | | | |
| Qwen1.5 (Bai et al., 2023) | 4B | 59.93 | 52.01 | 60.73 | 34.73 | 29.38 | 44.79 | 67.01 | 75.57 | 3.49 | 74.61 |
| w/ COALITION (ours) | 4B | **62.19** | **54.11** | **62.10** | **36.94** | **30.33** | **45.83** | **70.62** | **77.18** | **4.12** | **75.14** |
| Qwen1.5 (Bai et al., 2023) | 7B | 69.94 | 61.06 | 80.35 | 50.94 | 40.51 | 57.35 | 69.53 | 79.54 | 57.01 | 81.00 |
| w/ COALITION (ours) | 7B | **71.27** | **62.86** | **82.19** | **53.11** | **42.48** | **58.91** | **70.87** | **81.29** | **59.36** | **83.14** |
| Qwen1.5 (Bai et al., 2023) | 14B | 78.78 | 65.57 | 85.98 | 60.49 | 51.53 | 68.99 | 76.01 | 81.45 | 69.37 | 84.19 |
| w/ COALITION (ours) | 14B | **84.26** | **69.91** | **88.48** | **63.14** | **53.15** | **71.28** | **80.31** | **83.48** | **72.08** | **86.39** |
| | | | | | | | | | | | |
| Mistral (Jiang et al., 2023a) | 7B | 59.60 | 64.78 | 84.26 | 57.42 | 41.98 | 59.71 | 74.43 | 81.66 | 48.52 | 69.21 |
| w/ COALITION (ours) | 7B | **65.08** | **67.42** | **87.76** | **59.01** | **44.79** | **62.89** | **76.92** | **85.01** | **53.35** | **74.02** |
| | | | | | | | | | | | |
| LLaMA3 (Dubey et al., 2024) | 8B | 62.23 | 60.14 | 80.13 | 50.17 | 26.81 | 43.89 | 73.24 | 79.54 | 50.11 | 68.96 |
| w/ COALITION (ours) | 8B | **69.85** | **61.35** | **83.57** | **56.07** | **38.38** | **54.95** | **75.17** | **83.10** | **78.85** | **79.00** |

Table 7: Analysis of **generality** of COALITION trained in a task-agnostic manner by performing SRO on general open-domain samples (instruction-answer pairs). The table summarises the accuracy achieved on general benchmarks comprising of 10 tasks from the open-llm leaderboard. Rationales generated by COALITION uniformly improves the performance on all the tasks for all the LMs belonging to different model families (Phi3, Qwen1.5, Mistral, Llama3) and varying parameter-scales (ranging from 4B to 14B).

In the above prompt templates, ⟨instruction⟩ is a placeholder for the actual task instruction $\mathcal{I}^{\mathcal{T}}$ and ⟨rationale⟩ is a placeholder for the rationale text.

## A.4 EVALUATION ON GENERAL BENCHMARKS VIA TASK-AGNOSTIC SRO

We measure the generality and effectiveness of COALITION on general benchmarks (comprising of 10 tasks) in the open-llm leaderboard by performing selective rationale optimisation (SRO) of the LLM in a task-agnostic manner on a randomly selected subset of CoT data comprising of general open-domain samples (instruction-answer pairs). Given a sample from the test-split of a dataset from the open-llm leaderboard, the LLM trained using COALITION is leveraged to obtain the rationales through the *generate* and the *refine* steps for evaluation. In particular, the generated rationale is appended to the prompt after the sample-instruction. Table 7 summarises the results for different LLM backbones where it can be seen that rationales generated using COALITION uniformly increases the performance on all the 10 tasks for LLMs belonging to all the model families and parameter-scales. This demonstrates the **generality** of COALITION framework to improve the performance on tasks even when the training (SRO) is performed on general-domain samples in a task-agnostic manner.

Notably, for Mistral-7B, there is a significant increase of $\sim 5.5\%$ on the MMLU task which measures the ability of the model to answer questions related to the world-knowledge. Similarly, there is an improvement of $\sim 3\%$ on truthful-QA, PIQA and close to $5\%$ improvement on GSM8K and CSQA. Likewise for Llama3-8B, there is a huge increase of more than $7\%$ on MMLU, $6\%$ on the Challenging version of the ARC task, $11 - 12\%$ on truthful-QA and similar improvements on the other tasks. Similar improvements are observed for the Qwen model at different parameter-scales.

## A.5 ADDITIONAL RELATED WORK

Mixture-of-Agents (Wang et al., 2024a) uses multiple open-source LLMs based agents to improve the output quality at inference-time by generating intermediate output simultaneously using each agent independently. Their framework comprises of multiple such layers of LLM agents such that the outputs generated by agents in a layer are fed to the LLM agents in the subsequent layer which are prompted to analyze the information in the responses generated by LLM agents in the previous layer. It is observed that the accuracy on several benchmarks improve by prompting multiple LLM agents in such a manner across multiple layers of agents. COALITION creates and uses multiple variants of same SLM to improve its ability to generate and refine rationales in a trainable manner without involving any external LLM.

Zephyr (Tunstall et al., 2023) rates the responses generated by Falcon (Penedo et al., 2023), Llama (Touvron et al., 2023), etc. using GPT-4 (Achiam et al., 2023) to tune Mistral-7B (Jiang et al., 2023a) via DPO. Legal liabilities and lack of transparency in the pre-training data of larger models restrict their use to train smaller models for commercial use. Some methods use mixture-

| Dataset | Rationales Generated by Best Baseline (SPIN) and LLM Variants of COALITIONfor Generate and Refine Steps |
|---------|---------------------------------------------------------------------------------------------------------|
| GSM8K | **Instruction:** John takes care of 10 dogs. Each dog takes .5 hours a day to walk and take care of their business. How many hours a week does he spend taking care of dogs?
**Best Baseline:** John spends 0.5 hours a day taking care of each dog, and he has 10 dogs. So, for one day, the total time spent is .5\*10=50. Since there are 7 days in a week, we multiply the daily total by 7, the total time spent is 7\*50=250.✗
**Generate:** John spends 0.5 hours a day for each of the 10 dogs, so for one day he spends .5\*10=5 hours. To find the time spent in a week, multiply by 7 days: 5\*7=25 hours a week.✗
**Refine:** Each dog requires 0.5 hours of care per day, and John has 10 dogs. So, the total time spent per day is .5\*10=5 hours. Since there are 7 days in a week, the total time spent per week is 5\*7=35 hours. ✓ |
| WinoGrande | **Instruction:** Terry tried to bake the eggplant in the toaster oven but the _______ was too big. A. eggplant, B. toaster
**Best Baseline:** The word that "was too big" is referring to the toaster because it is the object Terry was using to bake. Therefore, the correct answer is B. toaster. ✗
**Generate:** The sentence says Terry tried to bake the eggplant, but something was too big. The word "was too big" is referring to the toaster, as it couldn't fit the eggplant inside. So the answer is B. toaster. ✗
**Refine:** The sentence implies that Terry tried to bake the eggplant, but it "was too big" for the toaster oven. Since the eggplant is the item being placed into the toaster oven, and the issue is with its size, the correct answer is A. eggplant. ✓ |
| PIQA | **Instruction:** How to dry flowers? sol1: Find a dark, moist area with good circulation, such as an attic or unused closet. With unflavored dental floss, secure the bottom of the flowers' stems to a hanger so that they hang upside down to dry. Leave flowers for two to three weeks until completely dry, sol2: Find a dark, dry area with good circulation, such as an attic or unused closet. With unflavored dental floss, secure the bottom of the flowers' stems to a hanger so that they hang upside down to dry. Leave flowers for two to three weeks until completely dry.
**Best Baseline:** Sol 1 is correct because flowers need a dark and moist environment to dry properly. Moisture will help retain the natural beauty of the flowers. So, Solution 1 is the better method for drying flowers. ✗
**Generate:** Solution 1 mentions finding a dark, moist area, which could help preserve the flowers' freshness for a while before they dry. ✗
**Refine:** Solution 2 is the correct approach because it suggests finding a dark, dry area with good air circulation, which is ideal for drying flowers without mold or decay. ✓ |
| HellaSwag | **Instruction:** Then he takes a small stone from the flowing river and smashes it on another stone. He starts to crush the small stone to smaller pieces. He _______________. A. cuts the center stone in half and blow it on to make it bigger, B. grind it hard to make the pieces smaller, C. eventually brings it back into view and adds it to the smaller ones to make a small triangular shaped piece, D. starts to party with them and throw the pieces by hand while they celebrate.
**Best Baseline:** The best answer is A. cuts the center stone in half and blows it on to make it bigger because cutting the stone and blowing on it would magically enlarge the pieces. ✗
**Generate:** The correct answer is C. eventually brings it back into view and adds it to the smaller ones to make a small triangular-shaped piece. This suggests he's shaping the stone into smaller pieces and eventually creating a triangle, which aligns with the idea of breaking the stone down. ✗
**Refine:** The best answer is B. grinds it hard to make the pieces smaller. This accurately matches the action described in the scenario, where the small stone is being smashed and crushed into smaller pieces. The goal is to break it down, and "grinding it hard" reflects that process. ✓ |
| CSQA | **Instruction:** When learning about the world and different cultures, what is important if you are committed to eliminating preconceived notions. A. newness, B. loss of innocence, C. enlightenment, D. open mind, E. smartness
**Best Baseline:** The correct answer is A. newness because when you learn about different cultures, everything you encounter will feel new and unfamiliar. Newness is what eliminates preconceived notions. ✗
**Generate:** The best answer is C. enlightenment because as you learn more about different cultures, you become enlightened, which helps you overcome your preconceived ideas. ✗
**Refine:** The correct answer is D. open mind because if you're committed to eliminating preconceived notions, the most important factor is being receptive and willing to accept new ideas and perspectives. ✓ |

Table 8: Qualitative analysis and comparison of rationales generated by COALITION with the best baseline - SPIN on all the five datasets on the three task domains. COALITION yields better rationales using *generate* and *refine* steps by leveraging the distinct LLM variants.

of-experts where the task-instruction is routed to suitable LLM expert either at the query-level (Lu et al., 2024a), in latent space (Jiang et al., 2024a) or at the output-layer (Si et al., 2023). On similar lines, COALITION employs a controller that is trained using the preference data collected during the DPO training to select the LLM variants for *generate* and *refine* steps. COALITION creates labelled data to train the controller based on which LLM variants's rationale gets selected during selective rationale optimisation.

## A.6   QUALITATIVE ANALYSIS

Table 8 shows a qualitative comparison of rationales generated by COALITION with the best baseline - SPIN on all the five datasets on the three task domains. COALITION yields better rationales using *generate* and *refine* steps by leveraging the distinct LLM variants.

## A.7   HUMAN STUDY FOR RATIONALE EVALUATION

We conducted a human study to evaluate the effectiveness of rationales obtained using the proposed COALITION framework. The following steps describe creation of data for human evaluation:

**Dataset Creation for Human Evaluation**

1. We collected a total of 75 samples by taking an equal number of samples for each task i.e. 15 samples randomly from the test sets of each of the 5 task datasets.

2. For each sample, we obtain the rationales $R1_g$, $R2_g$ from the two LLM variants at the generate step. Based on the variant selected by the controller for the generate step, the corresponding generated rationale $R_g$ is considered for refinement.

3. The selected generated rationale $R_g$ is used by the controller to determine the variant that should be used to refine the selected generated rationale. Once the variant is selected, it is used to refine the selected generated rationale to obtain the refined rationale – $R_r$.

Once the above rationales are obtained, we employed two paid human annotators and presented them with the instruction in each sample along with different rationales obtained above. The human evaluators are asked to judge the quality of different rationales based on the following questions and guidelines:

**Questions and Guidelines**

1. Question 1: Is the final rationale obtained from COALITION useful for answering the question correctly? The rationale is useful if it is correct and provides the correct explanation on how the answer for the instruction in the sample should be derived. Provide a label out of 0 or 1 such that 0 means that the final rationale is totally wrong; and 1 means that the final rationale is totally correct.

2. Question 2: Compare the selected generated rationale $R_g$ with the refined rationale $R_r$ obtained after refining $R_g$. Provide a label of 0 or 1 where 1 means that the refinement improved the generated rationale and 0 means there was no improvement.

3. Question 3: Compare the two rationales obtained using the two variants at the generate step - $R1_g$ and $R2_g$. Provide a label of 0 or 1 where 0 means that none of the rationales is better than the other and 1 means that one rationale is better than the other.

4. Question 4: In Question 3, in case one rationale is better than the other (between the rationales obtained from two variants at generate step), select the better rationale.

**Definition of Metrics Estimated from Human Labels**

Different rationales were presented to human evaluators in jumbled order to avoid biases while comparing rationales. Based on the judgement labels provided by the human evaluators for 4 questions above for the 75 samples, we estimate the following metrics:

1. Final Rationale Alignment – % proportion of samples which were assigned label 1 i.e. totally correct.

2. Improvement using Refinement - % proportion of samples where the refined rationale $R_r$ was judged to be improving the generated rationale $R_g$.

3. Diversity b/w two Rationales from Generate Step - % proportion of samples where the two rationales $R1_g$ and $R2_g$ obtained from two variants at generate step are different i.e. cases where one of the two rationales is better than the other (label 1). This metric is estimated to verify if the variants truly generate distinct rationales.

4. Better Rationale Alignment with Likelihood based Selection: We consider samples where label 1 is provided to Question 3 i.e. one of the generated rationales is judged better than the other generated rationale (comparing $R1_g$ and $R2_g$). We estimate the metric as % proportion cases from these samples where better rationale determined using likelihood-based utility score matches the better rationale from human judgement.

**Human Study Results and Discussion**

We compute the above metrics using the 75 samples used for human evaluation. We report the average of metrics obtained for the two human evaluators in Table 9. We discuss following observations from the results in Table 9:

1. From Table 9, we can observe that the final rationale alignment is 87.33% which means that final rationale obtained from COALITION is reliable and aligns with human preferences.

2. Rationale refinement helps since refinement improved the generated rationales for 36% cases. Thus, obtaining better rationales through refinement would also enable accuracy improvement on the final tasks as observed in the paper.

3. Rationales from Two Variants are diverse: It is observed that for 62.67% cases, one rationale obtained at generate step was judged to be better than the other generated rationale.

| Metric Name | Value (in %) |
|---|---|
| Final Rationale Alignment | 87.33 |
| Improvement using Refinement | 36.0 |
| Diversity b/w two Rationales from Generate Step | 62.67 |
| Better Rationale Alignment with Likelihood-based Selection | 80.85 |

Table 9: Human study results summarizing values of different metrics evaluated using human labels. It is observed that for good proportion of cases, final rationale obtained from COALITION aligns with human preferences, refinement helps improving generated rationales, the rationales obtained from two variants are diverse and better rationale judged by humans matches with winner rationale selected using likelihood based utility score.

    This means that employing two variants of same LLM is useful to obtain distinct and diverse rationales which are useful to improve quality of preference data for DPO.

4. Likelihood based rationale selection aligns with human preferences: For 80.85% cases, better generated rationale determined based on human preferences matches the better rationale based on likelihood-based utility score. This shows that our choice of using likelihood of final GT answer for selecting winner rationale aligns with human preferences and is suitable to obtain the preference data.

**Inter-Annotator Agreement**: We also report the inter-annotator agreement by estimating the Cohen's kappa coefficient which is commonly used to measure agreement between two annotators. For the human study, following is the Cohen-kappa coefficient for questions used to estimate each metric:

Cohen-Kappa coefficient for Final Rationale Alignment: 0.7112
Cohen-Kappa coefficient for Improvement using Refinement: 0.4851
Cohen-Kappa coefficient for Diversity b/w two Rationales from Generate Step: 0.7331
Cohen-Kappa coefficient for Better Rationale Alignment with Likelihood based Selection: 0.5105

Following is mapping of cohen-kappa coefficient value ranges with interpretation:

$0 - 0.2$: Slight agreement
$0.21 - 0.4$: Fair agreement
$0.41 - 0.6$: Moderate agreement
$0.61 - 0.8$: Substantial agreement
$0.81 - 1.0$: Almost Perfect agreement

Based on the coefficient obtained for different metrics and the above scale, it can be seen that human labels for final rationale alignment (0.7112) and diversity b/w rationales (0.7331) have substantial agreement while human labels for improvement using refinement (0.4851) and better rationale alignment with likelihood based selection (0.5105) have moderate agreement.

### A.8 RATIONALE EVALUATION USING LLM-AS-A-JUDGE

We perform the same evaluation as done for human study but instead of human evaluators, we use GPT-4o as the judge. GPT-4o is prompted with questions as used for human study for all the samples in the test split of each task dataset. Table 10 summarizes the values of metrics obtained using GPT-4o as judge where we report combined as well as dataset-wise metrics also since the number of samples for each dataset evaluated using GPT-4o is large.

It can be seen that using GPT-4o-as-a-judge yields similar (even more profound) trends as were observed from human study where quality of the final rationale obtained from COALITION is judged to be good for majority cases (for 82.55% samples on average) and refinement improves rationale quality (for ~60% cases on average). Further, the rationales obtained from LLM variants are diverse (for 71.21% cases on average) such that better rationale judged by GPT-4o aligns with winner rationale determined using likelihood-based utility score (for 88% cases on average).

| Metric Name | Combined across Tasks | GSM8K | WinoGrande | PIQA | HellaSwag | CSQA |
|---|---|---|---|---|---|---|
| Final Rationale Alignment | 82.55 | 77.69 | 77.83 | 85.29 | 83.40 | 88.53 |
| Improvement using Refinement | 59.66 | 69.19 | 61.29 | 57.33 | 53.47 | 57.01 |
| Diversity b/w two Rationales from Generate Step | 71.21 | 80.18 | 72.24 | 74.27 | 61.21 | 68.13 |
| Better Rationale Alignment with Likelihood-based Selection | 88.01 | 92.71 | 85.11 | 88.29 | 89.41 | 85.20 |

Table 10: LLM-as-a-judge results for evaluating the rationales using GPT-4o as a judge to estimate (i) Final Rationale Alignment, (ii) Improvement using Refinement, (iii) Diversity b/w two Rationales from Generate Step, and (iv) Better Rationale Alignment with Likelihood-based Selection.

| Method | GSM8K | WinoGrande | PIQA | HellaSwag | CSQA |
|---|---|---|---|---|---|
| COALITION w 2 LLM Variants | 81.06 | 77.13 | 83.26 | 63.23 | 82.06 |
| **COALITION w 3 LLM Variants** | **83.41** | **79.58** | **85.24** | **65.48** | **83.35** |

Table 11: Comparison of results by employing 3 LLM variants vs. 2 LLM variants (as done in main paper) in COALITION. Employing more variants improves the accuracy further.

## A.9 VARYING NUMBER OF LLM VARIANTS IN COALITION

The number of LLM variants is a hyper-parameter. We experimented with 2 LLM variants in the paper. As an ablation study, we perform an experiment where we employ and train three LLM variants and compare the accuracy with 2 LLM variants in table 11. It is observed that the accuracy on all the tasks improve uniformly with an average increase of 2% across different tasks. Thus, accuracy improvements over different baselines also get enhanced further with using 3 variants. We leave increasing the number of variants further to explore if it yields additional improvements as future work.

## A.10 AUTOMATED DIVERSITY ESTIMATION BETWEEN RATIONALES FROM TWO VARIANTS

To measure diversity between the rationales obtained from the two variants (for both generate as well as refine step), we estimate normalized lexical overlap between the rationales and take its complement as a measure of how distinct the rationales are. BLEU (Papineni et al., 2002) is commonly used metric in the NLP field to estimate overlap between two text sequences. Using Bleu, we estimate corresponding diversity metric i.e. Bleu-Diversity b/w rationales r1, r2 generated by the two variants respectively by taking complement of Bleu as follows:

Bleu-Diversity = 1 – Average[ Bleu(r1, r2)), Bleu(r2, r1) ]

**Note**: The values obtained using the overlap metric (BLEU) lie in the range of 0 to 1.

Table 12 shows the values of diversity metric for rationales obtained from two variants in COALITION for generate as well as refine steps respectively on all the tasks where it is observed that the diversity metric for all the tasks (for both generate and refine step) lie in the range of 0.68-0.80 (which is high on a scale of 0-to-1) which shows that the rationales obtained using the two variants are distinct from each other.

## A.11 ADDITIONAL MEASUREMENT OF RATIONALES USING PERPLEXITY

We conduct additional measurement of the rationales by estimating the perplexity of generating GT answer conditioned on rationales obtained at both generate and refine steps (for Llama-3-8B backbone). We also compare with the setting where no rationale is used. Lower perplexity means that training COALITION on winner/eliminated rationale using DPO enhances the LLM's confidence and chances of generating the correct answer. Table 13 summarizes the results where it is observed that using COALITION rationales reduces perplexity of GT answer. Also, using refined rationales results in lower perplexity compared to using rationales obtained from generate step highlighting the importance of refinement.

| Metric | GSM8K | WinoGrande | PIQA | HellaSwag | CSQA |
|---|---|---|---|---|---|
| Diversity b/w rationales obtained at Generate Step | 0.7525 | 0.7995 | 0.6893 | 0.8018 | 0.6827 |
| Diversity b/w rationales obtained at Refine Step | 0.7369 | 0.8048 | 0.6974 | 0.8149 | 0.7011 |

Table 12: Bleu-diversity metric b/w rationales from two variants for generate and refine steps respectively. Since Bleu overlap metric lies in range [0, 1], Bleu-diversity is also between 0-to-1. It can be seen that rationales from two variants are lexically diverse due to high value of the diversity metric for both generate and refine steps.

| Method | GSM8K | WinoGrande | PIQA | HellaSwag | CSQA |
|---|---|---|---|---|---|
| w/o any rationale | 11.29 | 6.92 | 6.73 | 8.47 | 8.84 |
| w COALITION rationales from Generate Step | 9.61 | 5.24 | 5.19 | 7.28 | 7.38 |
| **w COALITION rationales from Refine Step** | **8.47** | **4.48** | **4.46** | **5.37** | **6.53** |

Table 13: Perplexity (lower is better) of generating GT answer - (i) w/o any rationale, (ii) rationale from generate step in COALITION, and (iii) rationale from refine step in COALITION. It can be seen that COALITION's rationales reduces perplexity compared to not using any rationale. Further, refined rationales results in lower perplexity compared to using rationales from generate step.

## A.12 Number of Samples used to Train Variants

We report the number of samples used for each of the two variants to train them during the IFT stage as well as different iterations of DPO. During IFT, as discussed in implementation details section, a total of 180K samples were used. This IFT data was divided into two equal partitions such that 90K samples were used to train and obtain each LLM variant. IFT is performed in a task-agnostic manner. We summarize the number of training samples used for each variant during task-guided DPO in Table 14.

| Training Stage | GSM8K | WinoGrande | PIQA | HellaSwag | CSQA |
|---|---|---|---|---|---|
| DPO iteration-1 Generate Step | 1317 | 7297 | 2949 | 7649 | 1728 |
| DPO iteration-1 Refine Step | 1626 | 8934 | 3140 | 9795 | 1993 |
| DPO iteration-2 Generate Step | 1489 | 8379 | 3529 | 8029 | 1979 |
| DPO iteration-2 Refine Step | 1724 | 10093 | 3896 | 10764 | 2252 |

Table 14: Summary of number of samples used to train each variant for each DPO iteration for the generate and refine steps for different tasks.

