# OpenReview forum: "It Helps to Take a Second Opinion: Teaching Smaller LLMs To Deliberate Mutually via Selective Rationale Optimisation"
_ICLR.cc/2025/Conference — ICLR 2025 Poster_

### Official Review · Reviewer_Xpy3 · 2024-10-21

**Soundness:** 2
**Presentation:** 3
**Contribution:** 3
**Rating:** 6
**Confidence:** 4

**Summary:**

In the paper, COALITION is introduced to improve the performance of smaller LLMs in generating rationales through cross-communication. This is done by two variants of the same model fine-tuned in different ways.

**Strengths:**

- Presents COALITION framework, which refines smaller LLMs output by the same model
- Conduct experiments 6 LLMs ranging from 3B to 14B on 6 datasets
- Ablation study on components: distinct LLM variants, rationale selection and sample filtration

**Weaknesses:**

- One paper might be missing: Mixture-of-Agent (https://arxiv.org/abs/2406.04692) for cross-communication
- In line 196, I believe the utility score alone may not be sufficient in this case. A multi-dimensional evaluation, such as using LLM-as-judge, might be necessary. Additionally, the correctness of the refined generations is unclear, as LLMs often struggle with self-correction (https://openreview.net/forum?id=IkmD3fKBPQ, https://aclanthology.org/2024.findings-acl.826/).

**Questions:**

- In lines 77–78, you mention not relying on "external" LLMs. However, in lines 85–86, you refer to two distinct variants of SLM, which are trained differently. From my perspective, they are essentially no longer the same model.
- For evaluation, I think one important baseline is missing: self-refine (Madaan et al. 2023, https://openreview.net/forum?id=S37hOerQLB), which uses self-generated feedback to refine the rationales iteratively.

---

> ### Author Response · Authors · 2024-11-22
> **Author Response to Reviewer Xpy3 (1/4)**
>
> We would like to thank you for your encouraging review and feedback. We address your questions below:
>
> ---
> > One paper might be missing: Mixture-of-Agent (https://arxiv.org/abs/2406.04692) for cross-communication
> ---
>
> Thank you very much for suggesting the paper, it is a very interesting read and we have added the following discussion to related work (section 2, lines 130-133) in the modified main paper pdf. We have also added an elaborate discussion to the appendix A.5 in the modified pdf.
>
> **Main Paper (Section 2 Related work lines 130-133)**: Mixture-of-Agents [1] employs multiple open-source LLMs based agents and comprises of multiple layers of such LLM agents such that responses generated by agents in a layer are fed to LLM agents in the subsequent layer to refine the output.
>
> **Appendix (A.5 in the modified pdf draft)**: Mixture-of-Agents [1] uses multiple open-source LLMs based agents to improve the output quality at inference-time by generating intermediate output simultaneously using each agent independently. Their framework comprises of multiple such layers of LLM agents such that the outputs generated by agents in a layer are fed to the LLM agents in the subsequent layer which are prompted to analyze the information in the responses generated by LLM agents in the previous layer. It is observed that the accuracy on several benchmarks improve by prompting multiple LLM agents in such a manner across multiple layers of agents. COALITION creates and uses multiple variants of same SLM to improve its ability to generate and refine rationales in a trainable manner without involving any external LLM.
>
> [1] Wang, Junlin, Jue Wang, Ben Athiwaratkun, Ce Zhang, and James Zou. "Mixture-of-Agents Enhances Large Language Model Capabilities." arXiv preprint arXiv:2406.04692 (2024).
>
> ---
> > In line 196, I believe the utility score alone may not be sufficient in this case. A multi-dimensional evaluation, such as using LLM-as-judge, might be necessary.
> ---
>
> **Ablation on using LLM-as-judge for rationale selection** instead of likelihood based utility score was performed and reported in the original paper (Section 4.4, Table 5, rows 2 & 4 vs. COALITION - last row for Llama-3-8B backbone). We re-iterate the findings here for better understanding:
>
> The table below shows the comparison where it is observed that using likelihood of GT answer-based utility score to choose the winner/eliminated rationales gives better results as compared to using LLM-as-a-judge (Table 5, row 2 in paper) to rate the rationales for selection during DPO. Further, Table 5 - row 4 in paper corresponds to only using LLM-as-a-judge to rate rationales without using likelihood-based sample filtration (as discussed in section 3.2: lines 294-303) for removing noisy samples during DPO training. It was observed that just using LLM-as-a-judge without likelihood-based sample filtration results in even degraded accuracy. Following table summarizes these findings:
>
> |Method|GSM8K|WinoGrande|PIQA|HellaSwag|CSQA|
> |---|---|---|---|---|---|
> |COALITION w LLM-as-a-judge to rate rationales w likelihood based sample filtration|78.24|75.69|80.14|60.21|77.49|
> |COALITION w LLM-as-a-judge to rate rationales w/o likelihood-based sample filtration|73.19|71.37|76.16|56.92|77.01|
> |**COALITION w likelihood of GT answer to rate rationales and sample filtration**|**81.06**|**77.13**|**83.26**|**63.23**|**82.06**|
> ---
>
> Note: Here, we employed Llama-3-8B-IFT as the LLM-as-a-judge (since the aim is to improve SLM without using any external/larger LLM) using the same prompt and rating scale as used in the original paper [2]. The above results indicate that LLM-as-a-judge to rate rationales does not work effectively for smaller-scale LLMs.
>
> Further, we conducted a **human study** for evaluating the quality of rationales from COALITION as well as perform evaluation of rationales using **GPT-4o as judge** where we found that likelihood based utility score aligns well with human preferences. We describe them in the subsequent comments (3/4 and 4/4) of author response and encourage the reviewer to review the human study.
>
> [2] Zheng, Lianmin, Wei-Lin Chiang, Ying Sheng, Siyuan Zhuang, Zhanghao Wu, Yonghao Zhuang, Zi Lin et al. "Judging llm-as-a-judge with mt-bench and chatbot arena." Advances in Neural Information Processing Systems 36 (2023).
>
> ---
>
> We address the remaining concerns in the next comments of the author response ... (to be continued)

---

> > ### Author Response · Authors · 2024-11-22
> > **Contd. Author Response to Reviewer Xpy3 (2/4)**
> >
> > ---
> > > Additionally, the correctness of the refined generations is unclear, as LLMs often struggle with self-correction (https://openreview.net/forum?id=IkmD3fKBPQ, https://aclanthology.org/2024.findings-acl.826/).
> > ---
> >
> > Evaluations of COALITION in the paper concur with the observations from previous works that LLMs struggle with self-correction since it can be seen that self-refining (generate and refine using the same LLM variant) is not the preferred mode for controller in COALITION as discussed in section 4.3, figure 4. This can be supported and inferred through multiple experiments as reported and discussed in the paper as follows:
> >
> > * Controller-based generation and refinement of rationales (Section 4.3): Best accuracy is achieved in the setting when the controller is employed where it is observed that the controller prefers cross-refinement i.e. generates rationale using one LLM variant and refine the rationale using the other LLM variant (as shown through radar plot figure – 4). For about 65-75% of samples in the test-split across different tasks and datasets, the final rationale is obtained through cross refinement where the controller dynamically selects one LLM variant to generate rationale and the other variant to refine it conditioned on the input sample.
> > * Further, table 4 (section 4.4) shows that generating rationale using one LLM variant and cross refining using the other variant (through a fixed order of variants to generate and refine) gives much better accuracy (rows 5-6) vs. self-refining (rows 3-4) as well as not refining (rows 1-2).
> >
> > Further, based on the human study, it was observed that refinement in COALITION helps improving rationale quality as discussed in the subsequent author response comments (comment 3/4 and 4/4).
> >
> > ---
> > > For evaluation, I think one important baseline is missing: self-refine (Madaan et al. 2023, https://openreview.net/forum?id=S37hOerQLB), which uses self-generated feedback to refine the rationales iteratively.
> > ---
> >
> > Thanks for the suggestion. We evaluated the suggested Self-Refine baseline [3] (using Llama-3-8B backbone) where the LLM provides feedback to itself to refine the output. We have added it to the updated paper draft pdf in Table 1 (lines 385-386). We show the comparison of this baseline with COALITION in the table below also where it is observed that COALITION performs better than the baseline across different tasks. The discussion on comparison has been added to the updated draft of the pdf (in section 4.1, lines 430-431 and line 373).
> >
> > |Method|GSM8K|WinoGrande|PIQA|HellaSwag|CSQA|
> > |---|---|---|---|---|---|
> > |Self-Refine [3]|77.26|72.81|79.49|60.48|78.22|
> > |COALITION (w 2 LLM variants)|81.06|77.13|83.26|63.23|82.06|
> > |COALITION (w 3 LLM Variants)|83.41|79.58|85.24|65.48|83.35|
> > ---
> > Note: We also report results for experiment on COALITION w 3 LLM variants (as suggested by another reviewer) to compare enhanced accuracy gains over the suggested baseline.
> >
> > [3] Madaan, A., Tandon, N., Gupta, P., Hallinan, S., Gao, L., Wiegreffe, S., Alon, U., Dziri, N., Prabhumoye, S., Yang, Y. and Gupta, S., 2024. Self-refine: Iterative refinement with self-feedback. Advances in Neural Information Processing Systems, 36.
> >
> > ---
> >
> > > In lines 77–78, you mention not relying on "external" LLMs. However, in lines 85–86, you refer to two distinct variants of SLM, which are trained differently. From my perspective, they are essentially no longer the same model.
> >
> > ---
> >
> > COALITION improves reasoning ability of a smaller LM without using data or feedback from any external LLM. This is because COALITION begins with a single backbone (smaller-scale) LLM only that needs to be improved. It is as a part of its design where the proposed COALITION framework creates two distinct variants of the backbone LLM to obtain multiple rationales that can be used for preference optimization. The variants are obtained by creating two clones of the backbone LLM which are optimized on different data splits (during IFT and DPO stages) as part of its design. The variants are an internal part of the same system/framework since they are obtained using the same backbone LM that we started with. Therefore, COALITION does not use data or any feedback from an LLM that is external to the system or the framework to improve the backbone LLM. Hence, we show the effectiveness of the improved rationale generated by COALITION as a whole framework by itself which does not rely on external LLMs to achieve the improvements.
> >
> > ---
> >
> > We now discuss (in comments 3/4 and 4/4) the results of evaluating the rationales from COALITION through human study as well as GPT-4o as judge where we observed that final rationale obtained from COALITION is judged to be good for majority cases and refinement improves rationale quality. Further, the rationales obtained from LLM variants are diverse such that better rationale judged by humans/GPT-4o aligns with winner rationale determined using likelihood-based utility score ... (to be continued)

---

> > > ### Author Response · Authors · 2024-11-22
> > > **Contd. Author Response to Reviewer Xpy3 (3/4) - Human Study**
> > >
> > > **Human Study of Rationales**: We conducted a human study to evaluate the effectiveness of rationales obtained using the proposed COALITION framework. We discuss it here and have also added it to Appendix A.7 in the modified paper draft pdf. The following steps describes **creation of data for human evaluation**:
> > >
> > > * We collected a total of 75 samples by taking an equal number of samples i.e. 15 samples randomly from the test sets of each of the 5 task datasets.
> > > * For each sample, we obtain the rationales R1_g, R2_g from the two LLM variants at the generate step. Based on the variant selected by the controller for the generate step, the corresponding generated rationale is considered for refinement.
> > > * The selected generated rationale R_g is used by the controller to determine the variant that should be used to refine the selected generated rationale. Once the variant is selected, it is used to refine the selected generated rationale to obtain the refined rationale – R_r.
> > >
> > > Once the above rationales are obtained, we employed **two paid human annotators** and presented them with the instruction in each sample along with different rationales obtained above. The human evaluators are asked to judge the quality of different rationales based on the following **questions and guidelines**:
> > >
> > > * Question 1: Is the final rationale obtained from COALITION useful for answering the question correctly? The rationale is useful if it is correct and provides the correct explanation on how the answer for the instruction in the sample should be derived. Provide a label out of 0 or 1 such that 0 means that the final rationale is totally wrong; and 1 means that the final rationale is totally correct.
> > > * Question 2: Compare the selected generated rationale R_g with the refined rationale R_r obtained after refining R_g. Provide a label of 0 or 1 where 1 means that the refinement improved the generated rationale and 0 means there was no improvement.
> > > * Question 3: Compare the two rationales obtained using the two variants at the generate step - R1_g and R2_g. Provide a label of 0 or 1 where 0 means that none of the rationales is better than the other and 1 means that one rationale is better than the other.
> > > * Question 4: In Question 3, in case one rationale is better than the other (between the rationales obtained from two variants at generate step), select the better rationale.
> > >
> > > Different rationales were presented to human evaluators in jumbled order to avoid biases while comparing rationales. Based on the judgement labels provided by the human evaluators, we estimate the following **metrics**:
> > >
> > > * Final Rationale Alignment – % proportion of samples which were assigned label 1 i.e. totally correct.
> > > * Improvement using Refinement - % proportion of samples where the refined rationale R_r was judged to be improving the generated rationale R_g.
> > > * Diversity b/w two Rationales from Generate Step - % proportion of samples where the two rationales R1_g and R2_g obtained from two variants at generate step are different i.e. cases where one of the two rationales is better than the other (label 1). This metric is estimated to verify if the variants truly generate distinct rationales.
> > > * Better Rationale Alignment with Likelihood based Selection: We consider samples where label 1 is provided to Question 3 i.e. one of the generated rationales is judged better than the other generated rationale (comparing R1_g and R2_g). We estimate the metric as % proportion cases from these samples where better rationale determined using likelihood-based utility score matches the better rationale from human judgement.
> > >
> > > We compute the above metrics using the 75 samples used for human evaluation. We report the average of metrics obtained for the two human evaluators in the next comment of continued author response (4/4) ... (to be continued)

---

> > > > ### Author Response · Authors · 2024-11-22
> > > > **Contd. Author Response to Reviewer Xpy3 (4/4) - Results of Human Study and Evaluation using GPT-4o as judge**
> > > >
> > > > Values of different metrics from Human Evaluation:
> > > >
> > > > |Metric|Value (in %)|
> > > > |---|---|
> > > > |Final Rationale Alignment|87.33|
> > > > |Improvement using Refinement|36.0|
> > > > |Diversity b/w two Rationales from Generate Step|62.67|
> > > > |Better Rationale Alignment with Likelihood based Selection|80.85|
> > > >
> > > > ---
> > > >
> > > > Observations from above results:
> > > >
> > > > * It is seen that the final rationale alignment is 87.33% which means that final rationale obtained from COALITION is reliable and aligns with human preferences.
> > > > * Rationale refinement helps since refinement improved the generated rationales for 36% cases. Thus, obtaining better rationales through refinement would also enable accuracy improvement on the final tasks as observed in the paper.
> > > > * Rationales from Two Variants are diverse: It is observed that for 62.67% cases, one rationale obtained at generate step was judged to be better than the other generated rationale. This means that employing two variants of same LLM is useful to obtain distinct and diverse rationales which are useful to improve quality of preference data for DPO.
> > > > * Likelihood based rationale selection aligns with human preferences: For 80.85% cases, better generated rationale determined based on human preferences matches the better rationale based on likelihood-based utility score. This shows that our choice of using likelihood of final GT answer for selecting winner rationale aligns with human preferences and is suitable to obtain the preference data.
> > > >
> > > > ---
> > > >
> > > > **Inter-Annotator Agreement**: We also report the inter-annotator agreement by estimating the Cohen’s kappa coefficient which is commonly used to measure agreement between two annotators. For the human study, following is the Cohen-kappa coefficient for each question:
> > > >
> > > > * Inter-annotator agreement coefficient for Final Rationale Alignment: 0.7112
> > > > * Inter-annotator agreement coefficient for Improvement using Refinement: 0.4851
> > > > * Inter-annotator agreement coefficient for Diversity b/w two Rationales from Generate Step: 0.7331
> > > > * Inter-annotator agreement coefficient for Better Rationale Alignment with Likelihood based Selection: 0.5105
> > > >
> > > > Following is mapping of cohen kappa coefficient value ranges with interpretation:
> > > >
> > > > * 0 – 0.2: Slight agreement
> > > > * 0.21 - 0.4: Fair agreement
> > > > * 0.41 - 0.6: Moderate agreement
> > > > * 0.61 - 0.8: Substantial agreement
> > > > * 0.81 - 1.0: Almost Perfect agreement
> > > >
> > > > Based on the coefficient obtained for different metrics and the above scale, it can be noticed that human labels for final rationale alignment (0.7112) and diversity b/w rationales (0.7331) have substantial agreement while human labels for improvement using refinement (0.4851) and better rationale alignment with likelihood based selection (0.5105) have moderate agreement.
> > > >
> > > > ---
> > > >
> > > > **Rationale Evaluation using LLM-as-Judge** (Added to Appendix A.8 in the updated paper draft pdf): We perform the same evaluation as done for human study but instead of human evaluators, we use GPT-4o as the judge. GPT-4o is prompted with the questions as used for human study for all the samples in the test split of each task dataset. Following table summarizes the values of metrics obtained using GPT-4o as judge where we report dataset-wise metrics also since the number of samples for each dataset evaluated using GPT-4o is large:
> > > >
> > > > |Metric|Combined Across Tasks|GSM8K|WinoGrande|PIQA|HellaSwag|CSQA|
> > > > |---|---|---|---|---|---|---|
> > > > |Final Rationale Alignment|82.55|77.69|77.83|85.29|83.40|88.53|
> > > > |Improvement using Refinement|59.66|69.19|61.29|57.33|53.47|57.01|
> > > > |Diversity b/w two Rationales from Generate Step|71.21|80.18|72.24|74.27|61.21|68.13|
> > > > |Better Rationale Alignment with Likelihood based Selection|88.01|92.71|85.11|88.29|89.41|85.20|
> > > >
> > > > ---
> > > >
> > > > It can be seen that using GPT-4o-as-a-judge yields similar (even more profound) trends as were observed from human study where quality of the final rationale obtained from COALITION is judged to be good for majority cases (for 82.55% samples on average) and refinement improves rationale quality (for ~60% cases on average). Further, the rationales obtained from LLM variants are diverse (for 71.21% cases on average) such that better rationale judged by GPT-4o aligns with winner rationale determined using likelihood-based utility score (for 88% cases on average).
> > > >
> > > > ---

---

> > > > > ### Comment · Reviewer_Xpy3 · 2024-11-26
> > > > >
> > > > > Thanks for providing the additional experiments. But I will maintain the score.

---

> > > > > > ### Author Response · Authors · 2024-11-26
> > > > > >
> > > > > > Thank you for reviewing the author responses and providing your encouraging and valuable feedback.

---

### Official Review · Reviewer_JqYh · 2024-11-02

**Soundness:** 3
**Presentation:** 3
**Contribution:** 3
**Rating:** 8
**Confidence:** 4

**Summary:**

[Edit] I have updated my score based on the authors' response.

This paper proposes a novel framework "COALITION" that can be used with smaller language models (<13B) to generate rationales that lead to higher accuracy. COALITION uses two variants of a language model (variant in terms of instruction-finetune training data), and employs self/cross-refinement + DPO to train them. Finally during inference, a controller model picks a rationale to be used between the ones generated by the two variants. The authors present strong empirical results and a thorough ablation study.

**Strengths:**

* Strong empirical results including (1) Varied, thorough and strong set of baselines show the numerical significance of COALITION, and (2) results with multiple base-LLMs such as Phi3, Qwen, Mistral, Llama3
* Useful ablation studies that demonstrate the usefulness of their proposed components.

**Weaknesses:**

* Section 3 presents difficulty in understanding the training of LLM variants - I suggest the authors to clearly explain the initial training data differences between LV1, LV2, and M_IFT. It is my understanding that M_IFT was trained with the entire IFT data, whereas LV1 and LV2 (before refinement) were trained with different splits of the IFT data to ensure that they exhibited different behaviours, but I invite the authors to correct me if this is wrong, and also add a simplified, non-mathematical summary of the same in Section 3.
* To clarify, each LLM variant was trained wth DPO with both (1) generated rationales raked by utility scores, and (2) refined rationales ranked by utility scores?
* Section 3.3: What is the purpose of training a controller C to rank the variants, when M_IFT was already performing that role with its utility scores?
* Since this paper depends heavily on rationales, I suggest the authors to perform a human study on the rationales (generated, refined, and final) to check their quality. I understand that this work is primarily focused on improvement of accuracy, however, a preliminary understanding of the quality of the rationales generated is essential to have trust in this system.
* Lastly, I suggest the authors to discuss the explainability aspect of their framework in the Ethics section.

**Questions:**

[covered in Weaknesses]

---

> ### Author Response · Authors · 2024-11-22
> **Author Response to Reviewer JqYh (1/3)**
>
> We would like to thank you for your encouraging review and feedback. We address your questions below:
>
> ---
> > Section 3 presents difficulty in understanding the training of LLM variants - I suggest the authors to clearly explain the initial training data differences between LV1, LV2, and M_IFT. It is my understanding that M_IFT was trained with the entire IFT data, whereas LV1 and LV2 (before refinement) were trained with different splits of the IFT data to ensure that they exhibited different behaviours, but I invite the authors to correct me if this is wrong, and also add a simplified, non-mathematical summary of the same in Section 3.
> ---
>
> Yes it is correct that M_IFT is obtained by training the base LLM on the entire IFT data. The variants LV_1 and LV_2 are trained by randomly dividing the IFT data mix into two equal partitions and assigning one partition to each variant for training. We have added this clarification to Section 3.1 in the updated pdf draft (lines 228-230).
>
> ---
> > To clarify, each LLM variant was trained wth DPO with both (1) generated rationales raked by utility scores, and (2) refined rationales ranked by utility scores?
> ---
>
> Yes, each variant is trained via DPO on both the 1) generated rationales ranked by utility score and 2) refined rationales ranked by utility score.
>
> ---
> > Section 3.3: What is the purpose of training a controller C to rank the variants, when M_IFT was already performing that role with its utility scores?
> ---
>
> Since the GT answer is available only during training time, we are able to use M_IFT only during training phase to estimate likelihood of generating the GT answer conditioned on the rationale in the input to rank the rationales obtained from each variant. However, since the ground-truth answer is not available during testing/inference phase, it is not possible to rank the rationales using likelihood of the GT answer to select the appropriate rationale. Hence, to mitigate this, the controller is trained (through classification) using cross-entropy loss to select the variant that generates the rationale with higher likelihood conditioned on the given sample instruction. Once trained, during test/inference phase, given a sample instruction, the controller is then used (for both generate and refine steps) to select the variants that should be used to obtain the rationale.
>
> ---
> > Lastly, I suggest the authors to discuss the explainability aspect of their framework in the Ethics section.
> ---
>
> Thanks for the suggestion to add discussion on explainability to Ethics section. We have added the following discussion on explainability to the Ethics Statement section (lines 552-563):
>
> LLMs are commonly used to generate the final answer to an input question/instruction for various NLP tasks. However, it was shown that eliciting the LLM to generate a rationale first followed by the final answer results in better accuracy. A rationale is a statement in natural language that describes the steps which are required to derive the answer, or an explanation about how the question/instruction needs to be approached to arrive at the right answer. The proposed COALITION framework improves the reasoning ability of (smaller) LLMs by improving their ability to generate better rationales. Since the rationales provide an explanation about why the LLM generated the final answer instead of just generating the final answer, the rationales can be used as a means of explainability while generating the answer to an input question/instruction. Further, since COALITION generate and refine multiple rationales using variants of the same LLM, the generated and the refined rationales can be compared to identify differences in their explanation and quality. The identified differences can provide further insights about what needs to be modified in the explanations.
>
> ---
> > Since this paper depends heavily on rationales, I suggest the authors to perform a human study on the rationales (generated, refined, and final) to check their quality. I understand that this work is primarily focused on improvement of accuracy, however, a preliminary understanding of the quality of the rationales generated is essential to have trust in this system.
> ---
>
> As suggested, we conducted a **human study** (explained in the next comment (2/3) of author response) to evaluate the quality of rationales from COALITION. We found that quality of the final rationale obtained from COALITION is judged to be good for majority cases and refinement improves rationale quality. Further, the rationales obtained from LLM variants are diverse such that better rationale judged based on human preferences aligns with winner rationale determined using likelihood-based utility score. We also conducted a similar study using GPT-4o-as-judge instead of humans where we observed similar (and even more profound) trends.
>
> We now explain results of human study & using GPT-4o as judge in the next comment (2/3) of author response ... (to be continued).

---

> > ### Author Response · Authors · 2024-11-22
> > **Contd. Author Response to Reviewer JqYh (2/3) - Human Study**
> >
> > **Human Study of Rationales**: We conducted a human study to evaluate the effectiveness of rationales obtained using the proposed COALITION framework. We discuss it here and have also added it to Appendix A.7 in the modified paper draft pdf. The following steps describes **creation of data for human evaluation**:
> >
> > * We collected a total of 75 samples by taking an equal number of samples i.e. 15 samples randomly from the test sets of each of the 5 task datasets.
> > * For each sample, we obtain the rationales R1_g, R2_g from the two LLM variants at the generate step. Based on the variant selected by the controller for the generate step, the corresponding generated rationale is considered for refinement.
> > * The selected generated rationale R_g is used by the controller to determine the variant that should be used to refine the selected generated rationale. Once the variant is selected, it is used to refine the selected generated rationale to obtain the refined rationale – R_r.
> >
> > Once the above rationales are obtained, we employed **two paid human annotators** and presented them with the instruction in each sample along with different rationales obtained above. The human evaluators are asked to judge the quality of different rationales based on the following **questions and guidelines**:
> >
> > * Question 1: Is the final rationale obtained from COALITION useful for answering the question correctly? The rationale is useful if it is correct and provides the correct explanation on how the answer for the instruction in the sample should be derived. Provide a label out of 0 or 1 such that 0 means that the final rationale is totally wrong; and 1 means that the final rationale is totally correct.
> > * Question 2: Compare the selected generated rationale R_g with the refined rationale R_r obtained after refining R_g. Provide a label of 0 or 1 where 1 means that the refinement improved the generated rationale and 0 means there was no improvement.
> > * Question 3: Compare the two rationales obtained using the two variants at the generate step - R1_g and R2_g. Provide a label of 0 or 1 where 0 means that none of the rationales is better than the other and 1 means that one rationale is better than the other.
> > * Question 4: In Question 3, in case one rationale is better than the other (between the rationales obtained from two variants at generate step), select the better rationale.
> >
> > Different rationales were presented to human evaluators in jumbled order to avoid biases while comparing rationales. Based on the judgement labels provided by the human evaluators, we estimate the following **metrics**:
> >
> > * Final Rationale Alignment – % proportion of samples which were assigned label 1 i.e. totally correct.
> > * Improvement using Refinement - % proportion of samples where the refined rationale R_r was judged to be improving the generated rationale R_g.
> > * Diversity b/w two Rationales from Generate Step - % proportion of samples where the two rationales R1_g and R2_g obtained from two variants at generate step are different i.e. cases where one of the two rationales is better than the other (label 1). This metric is estimated to verify if the variants truly generate distinct rationales.
> > * Better Rationale Alignment with Likelihood based Selection: We consider samples where label 1 is provided to Question 3 i.e. one of the generated rationales is judged better than the other generated rationale (comparing R1_g and R2_g). We estimate the metric as % proportion cases from these samples where better rationale determined using likelihood-based utility score matches the better rationale from human judgement.
> >
> > We compute the above metrics using the 75 samples used for human evaluation. We report the average of metrics obtained for the two human evaluators in the next comment of continued author response (3/3) ... (to be continued)

---

> > > ### Author Response · Authors · 2024-11-22
> > > **Contd. Author Response to Reviewer JqYh (3/3) - Results of Human Study and LLM-as-Judge Evaluation**
> > >
> > > Values of different metrics from Human Evaluation:
> > >
> > > |Metric|Value (in %)|
> > > |---|---|
> > > |Final Rationale Alignment|87.33|
> > > |Improvement using Refinement|36.0|
> > > |Diversity b/w two Rationales from Generate Step|62.67|
> > > |Better Rationale Alignment with Likelihood based Selection|80.85|
> > >
> > > ---
> > >
> > > Observations from above results:
> > >
> > > * It is seen that the final rationale alignment is 87.33% which means that final rationale obtained from COALITION is reliable and aligns with human preferences.
> > > * Rationale refinement helps since refinement improved the generated rationales for 36% cases. Thus, obtaining better rationales through refinement would also enable accuracy improvement on the final tasks as observed in the paper.
> > > * Rationales from Two Variants are diverse: It is observed that for 62.67% cases, one rationale obtained at generate step was judged to be better than the other generated rationale. This means that employing two variants of same LLM is useful to obtain distinct and diverse rationales which are useful to improve quality of preference data for DPO.
> > > * Likelihood based rationale selection aligns with human preferences: For 80.85% cases, better generated rationale determined based on human preferences matches the better rationale based on likelihood-based utility score. This shows that our choice of using likelihood of final GT answer for selecting winner rationale aligns with human preferences and is suitable to obtain the preference data.
> > >
> > > ---
> > >
> > > **Inter-Annotator Agreement**: We also report the inter-annotator agreement by estimating the Cohen’s kappa coefficient which is commonly used to measure agreement between two annotators. For the human study, following is the Cohen-kappa coefficient for each question:
> > >
> > > * Inter-annotator agreement coefficient for Final Rationale Alignment: 0.7112
> > > * Inter-annotator agreement coefficient for Improvement using Refinement: 0.4851
> > > * Inter-annotator agreement coefficient for Diversity b/w two Rationales from Generate Step: 0.7331
> > > * Inter-annotator agreement coefficient for Better Rationale Alignment with Likelihood based Selection: 0.5105
> > >
> > > Following is mapping of cohen kappa coefficient value ranges with interpretation:
> > >
> > > * 0 – 0.2: Slight agreement
> > > * 0.21 - 0.4: Fair agreement
> > > * 0.41 - 0.6: Moderate agreement
> > > * 0.61 - 0.8: Substantial agreement
> > > * 0.81 - 1.0: Almost Perfect agreement
> > >
> > > Based on the coefficient obtained for different metrics and the above scale, it can be noticed that human labels for final rationale alignment (0.7112) and diversity b/w rationales (0.7331) have substantial agreement while human labels for improvement using refinement (0.4851) and better rationale alignment with likelihood based selection (0.5105) have moderate agreement.
> > >
> > > ---
> > >
> > > **Rationale Evaluation using LLM-as-Judge** (Added to Appendix A.8 in the updated paper draft pdf): We perform the same evaluation as done for human study but instead of human evaluators, we use GPT-4o as the judge. GPT-4o is prompted with the questions as used for human study for all the samples in the test split of each task dataset. Following table summarizes the values of metrics obtained using GPT-4o as judge where we report dataset-wise metrics also since the number of samples for each dataset evaluated using GPT-4o is large:
> > >
> > > |Metric|Combined Across Tasks|GSM8K|WinoGrande|PIQA|HellaSwag|CSQA|
> > > |---|---|---|---|---|---|---|
> > > |Final Rationale Alignment|82.55|77.69|77.83|85.29|83.40|88.53|
> > > |Improvement using Refinement|59.66|69.19|61.29|57.33|53.47|57.01|
> > > |Diversity b/w two Rationales from Generate Step|71.21|80.18|72.24|74.27|61.21|68.13|
> > > |Better Rationale Alignment with Likelihood based Selection|88.01|92.71|85.11|88.29|89.41|85.20|
> > >
> > > ---
> > >
> > > It can be seen that using GPT-4o-as-a-judge yields similar (even more profound) trends as were observed from human study where quality of the final rationale obtained from COALITION is judged to be good for majority cases (for 82.55% samples on average) and refinement improves rationale quality (for ~60% cases on average). Further, the rationales obtained from LLM variants are diverse (for 71.21% cases on average) such that better rationale judged by GPT-4o aligns with winner rationale determined using likelihood-based utility score (for 88% cases on average).
> > >
> > > ---

---

> ### Author Response · Authors · 2024-11-24
> **Thank you for reviewing author responses and increasing the score**
>
> We would like to thank the reviewer for reviewing our responses, increasing the score and provding valuable and encouraging feedback.

---

### Official Review · Reviewer_FNsr · 2024-11-04

**Soundness:** 3
**Presentation:** 3
**Contribution:** 3
**Rating:** 8
**Confidence:** 3

**Summary:**

This paper introduces a COALITION framework to improve the rationale generated from smaller language models without more powerful teacher models. LLMs typically excel at generating complex rationale but have limitations in their cost. COALITION addresses the limitations of SLMs by enabling two variants of the same model to generate and refine diverse rationales, optimizing them using a Selective Rationale Optimization (SRO) process to maximize task performance.

Their experiments verify the effectiveness of their method on three base models on five datasets. The result shows that cross-communication between model variants can boost performance compared to self-refinement and other baselines by up to 5%.

**Strengths:**

The paper contributes to rationale generation by addressing a gap: improving the performance of smaller language models without relying on more powerful large language models. While existing work often emphasises knowledge distillation or prompt-based methods involving LLMs, this paper introduces COALITION, a unique framework that leverages two variants of the same model to collaboratively generate and refine rationales.

Their IFT trains two variants of an SLM on separate data splits to exhibit distinct behaviours in generating and refining rationales. During SRO, these variants independently generate and cross-refine rationales for a given task, with utility scores assigned based on the likelihood of producing correct answers. This scoring guides DPO in refining the models’ outputs. The controller, trained on preference data from DPO, dynamically selects the optimal variant for generation and refinement during inference. This coordinated approach generates a higher-quality rationale, boosting SLM performance without external supervision.

The writing is well structured, and their experiments consistently improve on different models and tasks compared with the baseline method.

**Weaknesses:**

The authors quantitatively analyse the quality of the rationales based on the final task accuracy, determining rationale effectiveness by how well it leads to correct answers. However, they did not focus on conducting human evaluations or employing LLMs as judges to assess rationales to provide more insight into the rationale coherence or human preferences.

**Questions:**

The two variants model seems to be the most important part of the framework. How effectively do the two variants trained on separate data splits capture genuinely diverse reasoning paths? Could these variants' differences be further quantified or analysed to understand better their distinct contributions, e.g., what is the minimum data we need to train the variants?

---

> ### Author Response · Authors · 2024-11-22
> **Author Response to Reviewer FNsr (1/3)**
>
> We would like to thank you for your encouraging review and feedback. We address your questions below:
>
> ---
> > The authors quantitatively analyse the quality of the rationales based on the final task accuracy, determining rationale effectiveness by how well it leads to correct answers. However, they did not focus on conducting human evaluations or employing LLMs as judges to assess rationales to provide more insight into the rationale coherence or human preferences.
> ---
>
> We conducted additional evaluation of rationales through human study as well as using LLM as judge. We describe both the studies below:
>
> **Human Study of Rationales**: We conducted a human study to evaluate the effectiveness of rationales obtained using the proposed COALITION framework. We discuss it here and have also added it to Appendix A.7 in the modified paper draft pdf. The following steps describes **creation of data for human evaluation**:
>
> * We collected a total of 75 samples by taking an equal number of samples i.e. 15 samples randomly from the test sets of each of the 5 task datasets.
> * For each sample, we obtain the rationales R1_g, R2_g from the two LLM variants at the generate step. Based on the variant selected by the controller for the generate step, the corresponding generated rationale is considered for refinement.
> * The selected generated rationale R_g is used by the controller to determine the variant that should be used to refine the selected generated rationale. Once the variant is selected, it is used to refine the selected generated rationale to obtain the refined rationale – R_r.
>
> Once the above rationales are obtained, we employed **two paid human annotators** and presented them with the instruction in each sample along with different rationales obtained above. The human evaluators are asked to judge the quality of different rationales based on the following **questions and guidelines**:
>
> * Question 1: Is the final rationale obtained from COALITION useful for answering the question correctly? The rationale is useful if it is correct and provides the correct explanation on how the answer for the instruction in the sample should be derived. Provide a label out of 0 or 1 such that 0 means that the final rationale is totally wrong; and 1 means that the final rationale is totally correct.
> * Question 2: Compare the selected generated rationale R_g with the refined rationale R_r obtained after refining R_g. Provide a label of 0 or 1 where 1 means that the refinement improved the generated rationale and 0 means there was no improvement.
> * Question 3: Compare the two rationales obtained using the two variants at the generate step - R1_g and R2_g. Provide a label of 0 or 1 where 0 means that none of the rationales is better than the other and 1 means that one rationale is better than the other.
> * Question 4: In Question 3, in case one rationale is better than the other (between the rationales obtained from two variants at generate step), select the better rationale.
>
> Different rationales were presented to human evaluators in jumbled order to avoid biases while comparing rationales. Based on the judgement labels provided by the human evaluators, we estimate the following **metrics**:
>
> * Final Rationale Alignment – % proportion of samples which were assigned label 1 i.e. totally correct.
> * Improvement using Refinement - % proportion of samples where the refined rationale R_r was judged to be improving the generated rationale R_g.
> * Diversity b/w two Rationales from Generate Step - % proportion of samples where the two rationales R1_g and R2_g obtained from two variants at generate step are different i.e. cases where one of the two rationales is better than the other (label 1). This metric is estimated to verify if the variants truly generate distinct rationales.
> * Better Rationale Alignment with Likelihood based Selection: We consider samples where label 1 is provided to Question 3 i.e. one of the generated rationales is judged better than the other generated rationale (comparing R1_g and R2_g). We estimate the metric as % proportion cases from these samples where better rationale determined using likelihood-based utility score matches the better rationale from human judgement.
>
> We compute the above metrics using the 75 samples used for human evaluation. We report the average of metrics obtained for the two human evaluators in the next comment of continued author response (2/3) ... (to be continued)

---

> > ### Author Response · Authors · 2024-11-22
> > **Contd. Author Response to Reviewer FNsr (2/3) - Results of Human Study and LLM-as-Judge Evaluation**
> >
> > Values of different metrics from Human Evaluation:
> >
> > |Metric|Value (in %)|
> > |---|---|
> > |Final Rationale Alignment|87.33|
> > |Improvement using Refinement|36.0|
> > |Diversity b/w two Rationales from Generate Step|62.67|
> > |Better Rationale Alignment with Likelihood based Selection|80.85|
> >
> > ---
> >
> > Observations from above results:
> >
> > * It is seen that the final rationale alignment is 87.33% which means that final rationale obtained from COALITION is reliable and aligns with human preferences.
> > * Rationale refinement helps since refinement improved the generated rationales for 36% cases. Thus, obtaining better rationales through refinement would also enable accuracy improvement on the final tasks as observed in the paper.
> > * Rationales from Two Variants are diverse: It is observed that for 62.67% cases, one rationale obtained at generate step was judged to be better than the other generated rationale. This means that employing two variants of same LLM is useful to obtain distinct and diverse rationales which are useful to improve quality of preference data for DPO.
> > * Likelihood based rationale selection aligns with human preferences: For 80.85% cases, better generated rationale determined based on human preferences matches the better rationale based on likelihood-based utility score. This shows that our choice of using likelihood of final GT answer for selecting winner rationale aligns with human preferences and is suitable to obtain the preference data.
> >
> > ---
> >
> > **Inter-Annotator Agreement**: We also report the inter-annotator agreement by estimating the Cohen’s kappa coefficient which is commonly used to measure agreement between two annotators. For the human study, following is the Cohen-kappa coefficient for each question:
> >
> > * Inter-annotator agreement coefficient for Final Rationale Alignment: 0.7112
> > * Inter-annotator agreement coefficient for Improvement using Refinement: 0.4851
> > * Inter-annotator agreement coefficient for Diversity b/w two Rationales from Generate Step: 0.7331
> > * Inter-annotator agreement coefficient for Better Rationale Alignment with Likelihood based Selection: 0.5105
> >
> > Following is mapping of cohen kappa coefficient value ranges with interpretation:
> >
> > * 0 – 0.2: Slight agreement
> > * 0.21 - 0.4: Fair agreement
> > * 0.41 - 0.6: Moderate agreement
> > * 0.61 - 0.8: Substantial agreement
> > * 0.81 - 1.0: Almost Perfect agreement
> >
> > Based on the coefficient obtained for different metrics and the above scale, it can be noticed that human labels for final rationale alignment (0.7112) and diversity b/w rationales (0.7331) have substantial agreement while human labels for improvement using refinement (0.4851) and better rationale alignment with likelihood based selection (0.5105) have moderate agreement.
> >
> > ---
> >
> > **Rationale Evaluation using LLM-as-Judge** (Added to Appendix A.8 in the updated paper draft pdf): We perform the same evaluation as done for human study but instead of human evaluators, we use GPT-4o as the judge. GPT-4o is prompted with the questions as used for human study for all the samples in the test split of each task dataset. Following table summarizes the values of metrics obtained using GPT-4o as judge where we report dataset-wise metrics also since the number of samples for each dataset evaluated using GPT-4o is large:
> >
> > |Metric|Combined Across Tasks|GSM8K|WinoGrande|PIQA|HellaSwag|CSQA|
> > |---|---|---|---|---|---|---|
> > |Final Rationale Alignment|82.55|77.69|77.83|85.29|83.40|88.53|
> > |Improvement using Refinement|59.66|69.19|61.29|57.33|53.47|57.01|
> > |Diversity b/w two Rationales from Generate Step|71.21|80.18|72.24|74.27|61.21|68.13|
> > |Better Rationale Alignment with Likelihood based Selection|88.01|92.71|85.11|88.29|89.41|85.20|
> >
> > ---
> >
> > It can be seen that using GPT-4o-as-a-judge yields similar (even more profound) trends as were observed from human study where quality of the final rationale obtained from COALITION is judged to be good for majority cases (for 82.55% samples on average) and refinement improves rationale quality (for ~60% cases on average). Further, the rationales obtained from LLM variants are diverse (for 71.21% cases on average) such that better rationale judged by GPT-4o aligns with winner rationale determined using likelihood-based utility score (for 88% cases on average).
> >
> > ---
> >
> > We address the remaining questions in the next comment (3/3) of the author response ... (to be continued)

---

> > > ### Author Response · Authors · 2024-11-22
> > > **Contd. Author Response to Reviewer FNsr (3/3) - Diversity of Rationales and Number of Training Samples for Variants**
> > >
> > > ---
> > >
> > > > The two variants model seems to be the most important part of the framework. How effectively do the two variants trained on separate data splits capture genuinely diverse reasoning paths? Could these variants' differences be further quantified or analysed to understand better their distinct contributions, e.g., what is the minimum data we need to train the variants?
> > >
> > > ---
> > >
> > > **Diversity of Rationales**: As discussed in the human study above, variants generate diverse rationales where often one rationale is better than the other such that creating such a preference data to train the LLM via DPO enables it to generate better rationales. Further, we propose an automated metric - Bleu-Diversity, to estimate diversity between rationales:
> > >
> > > To measure diversity between the rationales obtained from the two variants (for both generate as well as refine step), we estimate normalized lexical overlap between the rationales and take its compliment as a measure of how distinct the rationales are. BLEU is commonly used metric in the NLP field to estimate overlap between two text sequences. Using Bleu, we estimate corresponding diversity metric i.e. Bleu-Div b/w rationales r1, r2 generated by the two variants respectively by taking complement of Bleu as follows:
> > >
> > > Bleu-Div = 1 – Average[ Bleu(r1, r2)), Bleu(r2, r1) ]
> > >
> > > Note: The values obtained using the overlap metric (BLEU) lie in the range of 0 to 1.
> > >
> > > Following tables shows the values of diversity metric for rationales obtained from two variants in COALITION for generate as well as refine steps respectively on all the tasks where it is observed that the diversity metric for all the tasks (for both generate and refine step) lie in the range of ~0.68-0.80 (which is high on a scale of 0-to-1) which shows that the rationales obtained using the two variants are distinct from each other. We have added this discussion and results to Appendix A.10 in the updated pdf draft.
> > >
> > > |Metric|GSM8K|WinoGrande|PIQA|HellaSwag|CSQA|
> > > |---|---|---|---|---|---|
> > > |Diversity b/w rationales from variants at **Generate** Step |0.7525|0.7995|0.6893|0.8018|0.6827|
> > > |Diversity b/w rationales from variants at **Refine** Step |0.7369|0.8048|0.6974|0.8149|0.7011|
> > >
> > > ---
> > >
> > > **Number of Samples to Train Variants:** We report the number of samples used for each of the two variants to train them during the IFT stage as well different iterations of DPO. During IFT, as discussed in implementation details section, a total of 180K samples were used. This IFT data was divided into two equal partitions such that 90K samples were used to train and obtain each LLM variant. IFT is performed in a task-agnostic manner. We summarize the number of training samples used for each variant during task-guided DPO:
> > >
> > > |Training Stage|GSM8K|WinoGrande|PIQA|HellaSwag|CSQA|
> > > |---|---|---|---|---|---|
> > > |DPO iteration-1 Generate Step |1317|7297|2949|7649|1728|
> > > |DPO iteration-1 Refine Step |1626|8934|3140|9795|1993|
> > > |DPO iteration-2 Generate Step |1489|8379|3529|8029|1979|
> > > |DPO iteration-2 Refine Step |1724|10093|3896|10764|2252|
> > >
> > > ---
> > >
> > > We have added this discussion on number of samples and above statistics to Appendix A.12 in the updated pdf draft.
> > >
> > > ---

---

> > > > ### Comment · Reviewer_FNsr · 2024-11-25
> > > >
> > > > Thank you to the authors for the additional details. I will maintain my score.

---

> > > > > ### Author Response · Authors · 2024-11-25
> > > > >
> > > > > Thank you for reviewing our work and the author response and providing encouraging feedback.

---

### Official Review · Reviewer_6DbD · 2024-11-04

**Soundness:** 3
**Presentation:** 2
**Contribution:** 3
**Rating:** 6
**Confidence:** 5

**Summary:**

Training on rationale or chain-of-thoughts have been known to be effective at enhancing a smaller LM's ability, yet due to transparency, copyright issues, this method has a fundamental limitations. The authors propose Coalition, a method that trains multiple LMs on different data splits & constructs a CoT preference dataset by comparing the likelihood (of producing the correct answer by conditioning on the CoT candidates) from the different models. This is effective because when training a single model, it could not generate a diverse set of response candidates. Compared to prompting-based baselines or iterative self-training baselines, the proposed methods show stronger performance on 5 benchmarks.

**Strengths:**

The paper compares with multiple baselines (Table 1), show that their method works across various scales (Table 3), and is supported by ablation experiments that each component does make the method more effective (Table 4,5).

**Weaknesses:**

* In addition to extrinsic measurements, perhaps it would be great to include intrinsic measurements of how the preference data quality improves through (1) choosing a winner rationale and (2) refinement.

* It is not clear how the number of LLM variants are decided and how the dataset is divided to train each LLM variant. Considering that this is the first step of the overall pipeline, an ablation experiment of deciding the number of variants or protocols for dividing data splits seems crucial but is missing.

* Compared to using LLM-as-a-Judge or reward models to choose winner rationale/eliminated rationale, how much effective is using rationale-conditioned GT answer likelihood heuristic? There should also be an experiment for this as it is a trivial baseline.

**Questions:**

Please see the questions mentioned in weaknesses. I will change my scores accordingly based on the author's response!

---

> ### Author Response · Authors · 2024-11-22
> **Author Response to Reviewer 6DbD (1/3)**
>
> We would like to thank you for your encouraging review and feedback. We address your questions below:
>
> ---
> > It is not clear how the number of LLM variants are decided and how the dataset is divided to train each LLM variant. Considering that this is the first step of the overall pipeline, an ablation experiment of deciding the number of variants or protocols for dividing data splits seems crucial but is missing.
>
> ---
>
> **Employing more Variants Improves Accuracy**: The number of LLM variants is a hyper-parameter. We experimented with 2 LLM variants in the paper. As an ablation study, we perform an experiment where we employ and train three LLM variants and compare the accuracy with 2 LLM variants in the table below. It is observed that the **accuracy on all the tasks improve uniformly by using 3 variants with an average increase of 2% compared to using 2 variants**. Thus, accuracy improvements over different baselines also get enhanced further with using 3 variants. We leave increasing the number of variants further to explore if it yields additional improvements as future work. We have added this discussion and results in Appendix A.9 in the modified paper draft pdf.
>
> |Method|GSM8K|WinoGrande|PIQA|HellaSwag|CSQA|
> |---|---|---|---|---|---|
> |COALITION w 2 LLM Variants|81.06|77.13|83.26|63.23|82.06|
> |**COALITION w 3 LLM Variants**|**83.41**|**79.58**|**85.24**|**65.48**|**83.35**|
>
> ---
>
> **Data Splits for Variants**: Regarding deciding data splits for different LLM variants in the original paper, we divide the samples randomly into 2 equal partitions and assign one unique partition to each of the 2 variants. This is because we want to ensure that the variants exhibit different behavior (generate distinct output for the same instruction) which can be achieved by randomly dividing the samples between the variants in equal proportion. Training variants on different but equal data splits makes their weights different which ensures that they exhibit different behavior. Also, samples are divided in equal proportions so that each variant gets trained on similar amount of data to avoid any biases.
>
> Note: For the above experiment using 3 LLM variants, samples are divided into 3 equal partitions such that one unique partition is assigned randomly to each of the three LLM variants.
>
> ---
>
> > Compared to using LLM-as-a-Judge or reward models to choose winner rationale/eliminated rationale, how much effective is using rationale-conditioned GT answer likelihood heuristic? There should also be an experiment for this as it is a trivial baseline.
>
> ---
>
> We performed and reported the suggested ablation in the original draft (section 4.4 - Ablation Studies) in Table 5 (row 2) for Llama-3-8B backbone on using LLM-as-a-judge to choose winner/eliminated rationale. We report the numbers again in the table below where it is seen that using likelihood of GT answer-based utility score to choose the winner/eliminated rationales gives better results as compared to using LLM-as-a-judge to rate the rationales for selection during DPO. Further, we also reported another ablation (Table 5 - row 4) where only LLM-as-a-judge is used to rate rationales without using likelihood-based sample filtration for removing noisy samples during DPO training (as explained in section 3.2: lines 294-303) where it was observed that just using LLM-as-a-judge without likelihood-based sample filtration results in degraded accuracy.
>
> |Method|GSM8K|WinoGrande|PIQA|HellaSwag|CSQA|
> |---|---|---|---|---|---|
> |COALITION w LLM-as-a-judge to rate rationales w likelihood based sample filtration|78.24|75.69|80.14|60.21|77.49|
> |COALITION w LLM-as-a-judge to rate rationales w/o likelihood-based sample filtration|73.19|71.37|76.16|56.92|77.01|
> |**COALITION w likelihood of GT answer to rate rationales and sample filtration**|**81.06**|**77.13**|**83.26**|**63.23**|**82.06**|
> ---
>
> Note: Here, we employed Llama-3-8B-IFT as the LLM-as-a-judge (since the aim is to improve SLM without using any external/larger LLM) using the same prompt and rating scale as used in the original paper [1]. The above results indicate that LLM-as-a-judge does not work effectively for smaller-scale LLMs.
>
> [1] Zheng, Lianmin, Wei-Lin Chiang, Ying Sheng, Siyuan Zhuang, Zhanghao Wu, Yonghao Zhuang, Zi Lin et al. "Judging llm-as-a-judge with mt-bench and chatbot arena." Advances in Neural Information Processing Systems 36 (2023).
>
> ---
>
> > In addition to extrinsic measurements, perhaps it would be great to include intrinsic measurements of how the preference data quality improves through (1) choosing a winner rationale and (2) refinement.
>
> ---
>
> We carry out intrinsic measurements of how preference data improves by choosing winner rationale and refinement in two ways -
> * Human Study of Rationales
> * Perplexity of Correct Answer using COALITION Rationales
>
> We describe both of these in the next comment (2/3) of the author response ... (to be continued)

---

> > ### Author Response · Authors · 2024-11-22
> > **Contd. Author Response to Reviewer 6DbD (2/3) - Measurement of Rationale Quality and Human Study**
> >
> > ---
> > > In addition to extrinsic measurements, perhaps it would be great to include intrinsic measurements of how the preference data quality improves through (1) choosing a winner rationale and (2) refinement.
> > ---
> >
> > **Perplexity of GT using Rationales:** We measure perplexity of generating GT answer conditioned on rationales obtained at generate and refine steps (for Llama-3-8B backbone). We compare with setting where no rationale is used. Lower perplexity is better. Table below shows that COALITION's rationales from refine step has lowest perplexity indicating that training COALITION on winner/eliminated rationale using DPO and refinement enables us to obtain rationales which enhance LLM’s confidence of generating correct answer. We have added this discussion and results to Appendix A.11 in the updated pdf draft.
> >
> > |Method|GSM8K|WinoGrande|PIQA|HellaSwag|CSQA|
> > |---|---|---|---|---|---|
> > |w/o rationales|11.29|6.92|6.73|8.47|8.84|
> > |w COALITION rationales from Generate Step|9.61|5.24|5.19|7.28|7.38|
> > |**w COALITION rationales from Refine Step**|**8.47**|**4.48**|**4.46**|**5.37**|**6.53**|
> >
> > ---
> >
> > **Human Study of Rationales**: We conducted a human study to evaluate the effectiveness of rationales obtained by selecting winner/eliminated rationale and refinement using the proposed COALITION framework. We discuss it here and have also added it to Appendix A.7 in the modified paper draft pdf. The following steps describes creation of data for human evaluation:
> >
> > * We collected a total of 75 samples by taking an equal number of samples i.e. 15 samples randomly from the test sets of each of the 5 task datasets.
> > * For each sample, we obtain the rationales R1_g, R2_g from the two LLM variants at the generate step. Based on the variant selected by the controller for the generate step, the corresponding generated rationale is considered for refinement.
> > * The selected generated rationale R_g is used by the controller to determine the variant that should be used to refine the selected generated rationale. Once the variant is selected, it is used to refine the selected generated rationale to obtain the refined rationale – R_r.
> >
> > Once the above rationales are obtained, we employed two paid human annotators and presented them with the instruction in each sample along with different rationales obtained above. The human evaluators are asked to judge the quality of different rationales based on the following questions and guidelines:
> >
> > * Question 1: Is the final rationale obtained from COALITION useful for answering the question correctly? The rationale is useful if it is correct and provides the correct explanation on how the answer for the instruction in the sample should be derived. Provide a label out of 0 or 1 such that 0 means that the final rationale is totally wrong; and 1 means that the final rationale is totally correct.
> > * Question 2: Compare the selected generated rationale R_g with the refined rationale R_r obtained after refining R_g. Provide a label of 0 or 1 where 1 means that the refinement improved the generated rationale and 0 means there was no improvement.
> > * Question 3: Compare the two rationales obtained using the two variants at the generate step - R1_g and R2_g. Provide a label of 0 or 1 where 0 means that none of the rationales is better than the other and 1 means that one rationale is better than the other.
> > * Question 4: In Question 3, in case one rationale is better than the other (between the rationales obtained from two variants at generate step), select the better rationale.
> >
> > Different rationales were presented to human evaluators in jumbled order to avoid biases while comparing rationales. Based on the judgement labels provided by the human evaluators, we estimate the following metrics:
> >
> > * Final Rationale Alignment – % proportion of samples which were assigned label 1 i.e. totally correct.
> > * Improvement using Refinement - % proportion of samples where the refined rationale R_r was judged to be improving the generated rationale R_g.
> > * Diversity b/w two Rationales from Generate Step - % proportion of samples where the two rationales R1_g and R2_g obtained from two variants at generate step are different i.e. cases where one of the two rationales is better than the other (label 1). This metric is estimated to verify if the variants truly generate distinct rationales.
> > * Better Rationale Alignment with Likelihood based Selection: We consider samples where label 1 is provided to Question 3 i.e. one of the generated rationales is judged better than the other generated rationale (comparing R1_g and R2_g). We estimate the metric as % proportion cases from these samples where better rationale determined using likelihood-based utility score matches the better rationale from human judgement.
> >
> > We compute the above metrics using the 75 samples used for human evaluation. We report the average of metrics obtained for the two human evaluators in the next comment of author response (3/3) ... (to be continued)

---

> ### Author Response · Authors · 2024-11-22
> **Contd. Author Response to Reviewer 6DbD (3/3) - Human Study Results**
>
> Values of different metrics from Human Evaluation:
>
> |Metric|Value (in %)|
> |---|---|
> |Final Rationale Alignment|87.33|
> |Improvement using Refinement|36.0|
> |Diversity b/w two Rationales from Generate Step|62.67|
> |Better Rationale Alignment with Likelihood based Selection|80.85|
>
> ---
>
> Observations from above results:
>
> * It is seen that the final rationale alignment is 87.33% which means that final rationale obtained from COALITION is reliable and aligns with human preferences.
> * **Rationale refinement helps** since refinement improved the generated rationales for 36% cases. Thus, obtaining better rationales through refinement would also enable accuracy improvement on the final tasks as observed in the paper.
> * **Rationales from Two Variants are diverse**: It is observed that for 62.67% cases, one rationale obtained at generate step was judged to be better than the other generated rationale. This means that employing two variants of same LLM is useful to obtain distinct and diverse rationales which are useful to improve quality of preference data for DPO.
> * Likelihood based rationale selection aligns with human preferences: For 80.85% cases, better generated rationale determined based on human preferences matches the better rationale based on likelihood-based utility score. This shows that our choice of using likelihood of final GT answer for selecting winner rationale aligns with human preferences and is suitable to obtain the preference data.
>
> **Thus, since variants generate diverse rationales where often one rationale is better than the other, creating such a preference data to train the LLM via DPO enables it to generate better rationales. Further, it was observed through human study that refining the rationales helps**.
>
> ---
>
> **Inter-Annotator Agreement**: We also report the inter-annotator agreement by estimating the Cohen’s kappa coefficient which is commonly used to measure agreement between two annotators. For the human study, following is the Cohen-kappa coefficient for each question:
>
> * Inter-annotator agreement coefficient for Final Rationale Alignment: 0.7112
> * Inter-annotator agreement coefficient for Improvement using Refinement: 0.4851
> * Inter-annotator agreement coefficient for Diversity b/w two Rationales from Generate Step: 0.7331
> * Inter-annotator agreement coefficient for Better Rationale Alignment with Likelihood based Selection: 0.5105
>
> Following is mapping of cohen kappa coefficient value ranges with interpretation:
>
> * 0 – 0.2: Slight agreement
> * 0.21 - 0.4: Fair agreement
> * 0.41 - 0.6: Moderate agreement
> * 0.61 - 0.8: Substantial agreement
> * 0.81 - 1.0: Almost Perfect agreement
>
> Based on the coefficient obtained for different metrics and the above scale, it can be noticed that human labels for final rationale alignment (0.7112) and diversity b/w rationales (0.7331) have substantial agreement while human labels for improvement using refinement (0.4851) and better rationale alignment with likelihood based selection (0.5105) have moderate agreement.
>
> ---
>
> **Rationale Evaluation using LLM-as-Judge** (Added to Appendix A.8 in the updated paper draft pdf): We perform the same evaluation as done for human study but instead of human evaluators, we use GPT-4o as the judge. GPT-4o is prompted with the questions as used for human study for all the samples in the test split of each task dataset. Following table summarizes the values of metrics obtained using GPT-4o as judge where we report dataset-wise metrics also since the number of samples for each dataset evaluated using GPT-4o is large:
>
> |Metric|Combined Across Tasks|GSM8K|WinoGrande|PIQA|HellaSwag|CSQA|
> |---|---|---|---|---|---|---|
> |Final Rationale Alignment|82.55|77.69|77.83|85.29|83.40|88.53|
> |Improvement using Refinement|59.66|69.19|61.29|57.33|53.47|57.01|
> |Diversity b/w two Rationales from Generate Step|71.21|80.18|72.24|74.27|61.21|68.13|
> |Better Rationale Alignment with Likelihood based Selection|88.01|92.71|85.11|88.29|89.41|85.20|
>
> ---
>
> It can be seen that using GPT-4o-as-a-judge yields similar (even more profound) trends as were observed from human study where quality of the final rationale obtained from COALITION is judged to be good for majority cases (for 82.55% samples on average) and refinement improves rationale quality (for ~60% cases on average). Further, the rationales obtained from LLM variants are diverse (for 71.21% cases on average) such that better rationale judged by GPT-4o aligns with winner rationale determined using likelihood-based utility score (for 88% cases on average).
>
> ---

---

> > ### Comment · Reviewer_6DbD · 2024-11-25
> >
> > Thank you for the responses! Most of my questions have been resolved. I think using clustering algorithms to split the data and training the LMs could also be an effective alternative but this is outside of scope. I'll raise my score considering the author's hard work during the rebuttal.

---

> > > ### Author Response · Authors · 2024-11-25
> > > **Thanks for reviewing the author response and increasing the score.**
> > >
> > > We would like to thank the reviewer for reviewing the author response and increasing the score.

---

### Author Response · Authors · 2024-12-04
**General Author Comment**

Dear all,

We would like to thank you for the time invested in reviewing our work and the valuable feedback that has helped in further strengthning our submission. We have tried our best to clarify the doubts and have performed the requested evaluations that support the findings of our proposed approach. The experiments and human study showed that rationales from the proposed framework (COALITION) are useful, refinement of rationales help, rationales from variants are diverse and likelihood-based selection of winner rationale aligns with human preferences.

---

### Meta-Review · Area_Chair_qfqL · 2024-12-16

**Metareview:**

This paper introduces a framework (COALITION) to improve the rationale generated from smaller language models without more powerful teacher models. LLMs typically excel at generating complex rationale but have limitations in their cost. It addresses the limitations of smaller LMs by enabling two variants of the same model to generate and refine diverse rationales, optimizing them using a Selective Rationale Optimization process to maximize task performance.

The paper contributes to rationale generation by addressing a gap: improving the performance of smaller language models without relying on more powerful large language models. It presents strong empirical results including (1) Varied, thorough and strong set of baselines show the numerical significance of COALITION, and (2) results with multiple base-LLMs such as Phi3, Qwen, Mistral, Llama3.

There have been concerns regarding evaluation details, and requests for human studies, which the authors have clarified / added in their response.

**Additional Comments On Reviewer Discussion:**

There have been concerns regarding evaluation details, and requests for human studies, which the authors have clarified / added in their response.

---

### Decision · Program_Chairs · 2025-01-22

Accept (Poster)